# Soft-interfaced liquid crystal microfluidics can probe the rigidity of lipid vesicles
Cansu Dedeoglu [1] & Emre Bukusoglu [1,2] ✉

Liquid crystal (LC)-aqueous interfaces were shown to respond to the phospholipid interactions via optically observable ordering transitions; however, past attempts lack the quantification of the transport and fusion kinetics of the vesicles at the interfaces. Herein, we investigated the response of flowing LC-aqueous interfaces upon fusion of the vesicles formed by pure 1,2-dilauroyl-sn-glycero-3-phosphocholine (DLPC), 1,2-dioleoyl-sn-glycero-3-phosphocholine (DOPC), 1,2-dipalmitoyl-sn-glycero-3-phosphocholine (DPPC), or egg sphingomyelin or their mixtures with guest molecules. Using stabilized LC-aqueous interfaces in transparent microfluidic chips that allow spatiotemporal quantification using fluorescence, confocal, and polarized light microscopy, we demonstrated that flowing LC interfaces provide a rapid response to lipid adsorption, where their spatiotemporal interfacial distribution differs depending on the mechanical properties of their vesicles. We show that cholesterol-dissolved lipid complexes result in distinct LC-response kinetics, mainly associated to the changes in their rigidities. Considering the critical role of the mechanical properties of cell membranes in proper cellular function, this study is significant as it offers a continuous and rapid early diagnosis platform for detecting minor mechanical alterations in lipid bilayers, which may lead to cell dysfunction that contributes to critical diseases.

The plasma membranes of mammalian cells provide selective transport with their environment while facilitating communication, compartmentalization, intercellular interaction, and energy transduction[1]. Membrane integrity is crucial for cells since the membrane is very responsive to the changes in extracellular and intracellular environments, and disruption of the cell membrane can cause calcium toxicity, activation of proteolysis, osmotic stress, oxidative damage, and even cell death[2]. Membrane integrity is strongly dependent on its mechanical properties[3]. Studies showed that critical diseases in living organisms are related to changes in the mechanical properties of the cell membranes, for example, the toxic accumulation of psychosine is related to localized rigid areas of membranes and therefore, Krabbe's disease[4,5]. Membrane elasticity can be correlated with cell elasticity in some contexts, and cancer cells are often observed to be of lower elasticity, enhancing their migration and motility[6]. For example, melanin granules significantly alter the elasticity of the cells, which is an indicator of the metastatic ability of melanoma cells[7]. Cancer cells exhibit lower elasticity when compared with the normal, healthy cells, which has an impact on early diagnosis of cancers since cancer cells change and optimize their elasticity for migration during metastasis[6,8,9]. In the pathological studies of cancer, extracellular vesicles (EV) have been involved in recent years, and are known as "hallmarks of cancer"[10]. Exosomes and shed microvesicles are classes of

EVs, indicated as biomarkers and therapeutic tools[11,12] and they can be separated from different body fluids such as blood, urine, and cerebrospinal fluid[13]. These are referred to as "biocargos" that carry information about inflammatory responses, cell migration, invasion, and metastasis[14]. Some common guest molecules regulate the mechanical properties of the membrane, such as surfactants, proteins, cholesterol, and ceramides[15–18]. These guest molecules are naturally common in the cell membrane, and changes in their composition result in changes in the mechanical properties of the membrane. The most common molecule that regulates the mechanical properties of the cell membrane is cholesterol[19,20]. Although cholesterol is essential for the synthesis of bile acids and steroid hormones[21], its excess (or shortage) may influence membrane fluidity significantly, change membrane protein functions, and cause cell dysfunctions. Recently, it has been reported that excess cholesterol is related to some common diseases such as chronic kidney disease, Alzheimer's disease, and non-alcoholic fatty liver disease[22–25]. Overall, beyond the mechanistic characterizations of the cell membranes, their quantification is critically important in tracking diseases.

Liquid crystals (LC) are unique due to their long-range orientational ordering, fluidity, and optical birefringence. Thermotropic nematic phases of the LCs (for example, 4-cyano-4'-pentylbiphenyl (5CB)) are a widely studied phase due to the simple long-range single director orientation that is

[1]Department of Chemical Engineering, Middle East Technical University, Ankara, Türkiye. [2]Department of Micro and Nanotechnology, Middle East Technical University, Ankara, Türkiye. ✉e-mail: emrebuk@metu.edu.tr

sensitive to the external stimuli. Of particular interest in the thermotropic LCs is their stable, responsive interfaces with aqueous phases. Such interfaces, when considered with the elastic properties arising from their long-range ordering, present unique opportunities. For example, the physico-chemical properties of the LC-aqueous interfaces that were usually comparable to the cellular membranes make it a good model for studying the interactions and assemblies of the biologically relevant species at interfaces[26]. Further, the LC-aqueous interface is soft and deformable; molecules dissolved in the aqueous phase are mobile and able to transport to the interface, where they cause changes in the interfacial alignment of the 5CB mesogens[27]. During such processes, the highly responsive LC-aqueous interfaces enable optical amplification of the interaction of the interfacial biomolecular species in real time[28]. Past studies showed that these properties of the LC-aqueous interfaces provide useful platforms to understand the coupling between the anisotropic ordering of LC and the organization of the interfacial species[29–31]. One of the most thoroughly investigated systems is the adsorbed layers of the lipids at the LC-aqueous interfaces, which were critically related to the rate of the transport of the lipids from their vesicle suspensions and their elastic interactions at the interfaces[32]. Building on these early findings, studies showed that disruption of the lateral organization of phospholipid monolayers at the stagnant LC-aqueous interfaces by protein-coated nanoparticles[33] or amyloid-forming peptides generates distinct changes in LC orientational ordering, enabling label-free detection and real-time response[34]. Subsequent studies showed that lipid-coated LC-aqueous interface can be directly utilized for probing protein-lipid interactions in a similar manner, allowing the highly sensitive detection of cytoplasmic proteins[35] and mutation-dependent activity of toxins, highlighting the importance of studying biomolecular interactions at the interface[36]. Building on this approach, LC droplets have also been used to investigate nanoscale lipid–protein interactions, allowing for quantitative analysis of lipid–protein coupling at submicrometer scales by producing orientational transitions[37,38]. In addition, lipid-coated LC droplets in aqueous media emerged as a platform for controlling director fields through the organization at the interface, further highlighting the characterization of lipids[39,40]. In parallel, phospholipid-coated LC shells were demonstrated to create stable lipid islands whose spatial organization induces a change in the director inside the shell, demonstrating how confinement and curvature can enhance interfacial lipid heterogeneity[41]. Moreover, lateral heterogeneity and redistribution of the lipids at the interface can be caused by LC phase transitions themselves, as demonstrated by the local concentration of phospholipids into high-density regions caused by smectic LC nucleation at aqueous interfaces[42].

In addition to lipids and amphiphilic biomolecular interactions-mediated ordering transitions, LC-aqueous interfaces have been extensively used as optical biosensing platforms, where adsorption of biological species such as glucose[43], polymeric surfactants[44], or reaction-generated products such as α-glucosidase to inhibit anti-diabetic drugs[45] induces changes in orientational order. These interfaces have also emerged in aptamer-functionalized systems that detect pesticides[46] and nucleic acid-linked devices, enabling antibiotic sensing[47]. Recent studies revealed that synthetic enzyme mimics, such as fullerene-based catalytic assemblies, can induce a change in the optical appearance of LC, further highlighting LC platforms in biosensing applications[48,49].

Synthetic phospholipids and their vesicles are used as mimic systems for cell membranes and the extracellular vesicles. The surfactants and ceramides usually present a monotonous effect on phospholipid bilayers; surfactants soften the bilayers by shifting the melting transition temperature to lower values[50], while ceramides make them stiffer and make the membrane more ordered[51,52]. However, cholesterol has a non-monotonous and non-universal effect on bilayers, which have been shown to be dependent on the chemical structure of the phospholipids, chain saturation, and phase of the lipid[20,53–56]. The effect of cholesterol on phospholipid bilayers can be characterized by measuring their mechanical properties, such as permeability, fluidity, and bending rigidity[53,57]. The commonly studied property is the bending rigidity ($K_c$), and it can be measured by using different methods

such as micropipette aspiration, neutron-spin echo spectroscopy, H-NMR spectroscopy, fluctuation spectroscopy, electrodeformation, and vesicle size distribution[58]. Measuring $K_c$ is crucial to understanding how cholesterol changes the stiffening of the bilayer. One of the striking results in the past studies is that cholesterol causes an increment in $K_c$ of the phospholipid that has fully saturated chains, but when there are two monounsaturated chains, there is no net trend in $K_c$[20]. 1,2-dioleoyl-sn-glycero-3-phosphocholine (DOPC) exhibited an increase in $K_c$ with increasing cholesterol amount, shown by size distribution analysis, H-NMR spectroscopy, and molecular dynamics simulations[55,56]. It is shown that the $K_c$ of Egg Sphingomyelin (Egg SM) decreased with the cholesterol amount measured by electrode formation and fluctuation spectroscopy[53]. On the other hand, $K_c$ of 1,2-dimyristoyl-sn-glycero-3-phosphocholine (DMPC) and 1-stearoyl-2-oleoyl-sn-glycero-3-phosphocholine (SOPC) increased with even small amounts of cholesterol ($x_c = 0.3$), while DOPC remained invariant of cholesterol content[54]. Another study demonstrated that the $K_c$ of 1,2-dipalmitoyl-sn-glycero-3-phosphocholine (DPPC) and N-(hexadecanoyl)-hexadecasphing-4-enine-1-phosphocholine (DPSM) increased up to 40% mol cholesterol, and 1-palmitoyl-2-oleoyl-sn-glycero-3-phos-phocholine (POPC) and DOPC decreased up to the same amount[59]. Their understanding gained from the literature shows that cholesterol has a non-universal behavior on the mechanical properties of the phospholipid membranes. Here, we aim to contribute to this literature by investigating the influence of the rigidity of the phospholipid vesicles on their fusion characteristics. For the characterization of the fluidity of the membranes with cholesterol, shear rheology of the lipid monolayers is commonly used. Binary mixtures of POPC with cholesterol or Egg SM with cholesterol showed enhancement of fluidity, with 30% cholesterol shown by the reduced viscosity with shear strain[60]. DPPC monolayers exhibited increasing surface dilatational viscosity with increasing cholesterol amount up to 80%[61].

The fusion of the vesicles to the LC-aqueous interface occurs via collision and spreading from the vesicles in the dispersion. The formed monolayers of lipids cause a change in the LC interfacial orientation (anchoring transition), resulting in a significant difference in their optical appearance under polarized microscopy. Resulting from the anchoring transition, the elasticity of the LC causes a phase separation of the adsorbed lipids into lipid-rich and lipid-lean domains at the interface[29]. These transport and phase separation processes lead LC-aqueous interfaces to provide characterization methods of the properties of lipids by measuring their diffusivity, fluidity, and localization by monitoring the optical changes occurring at the underlying LC phase[30]. To date, studies have predominantly been conducted in stagnant LC-water interfaces. However, recent studies showed that additional information regarding the mass transfer of the interfacial adsorbates at the elastic LC interfaces can be obtained when mass transfer was coupled with the flow[62,63]. More recently, studies showed that "flowing" LC-aqueous interfaces can be maintained in the microfluidic systems that are stable up to significant pressure differences and maintain responsive properties against interfacial adsorbates[64,65]. This continuous, responsive LC-aqueous interface system can provide a platform to study both homogenous and heterogeneous adsorbed layers, highlighting the potential for early diagnosis of diseases, tracking advanced release platforms, and high-throughput analytical platforms.

In this study, we investigated the fusion characteristics of synthetic or natural, biologically relevant phospholipid vesicles and their mixtures with common guest molecules such as cholesterol, surfactants, and lipids to the 5CB-aqueous interfaces. Utilizing the unique properties of LCs and their aqueous interfaces formed in microfluidic systems, we aim to provide a platform for understanding the role of the rigidity of the phospholipid vesicles on their fusion to the LC-aqueous interfaces under flow conditions. For this, we tracked the spatiotemporal changes at the LC-aqueous interfaces caused by the fusion of the lipid vesicles under both stagnant and continuous conditions to evaluate the role of the shear and flow conditions on such processes. The designed platform can provide detailed characteristics of the effect of cholesterol on phospholipid vesicles, highlighting a non-universal behavior. The findings of our study offer insights into

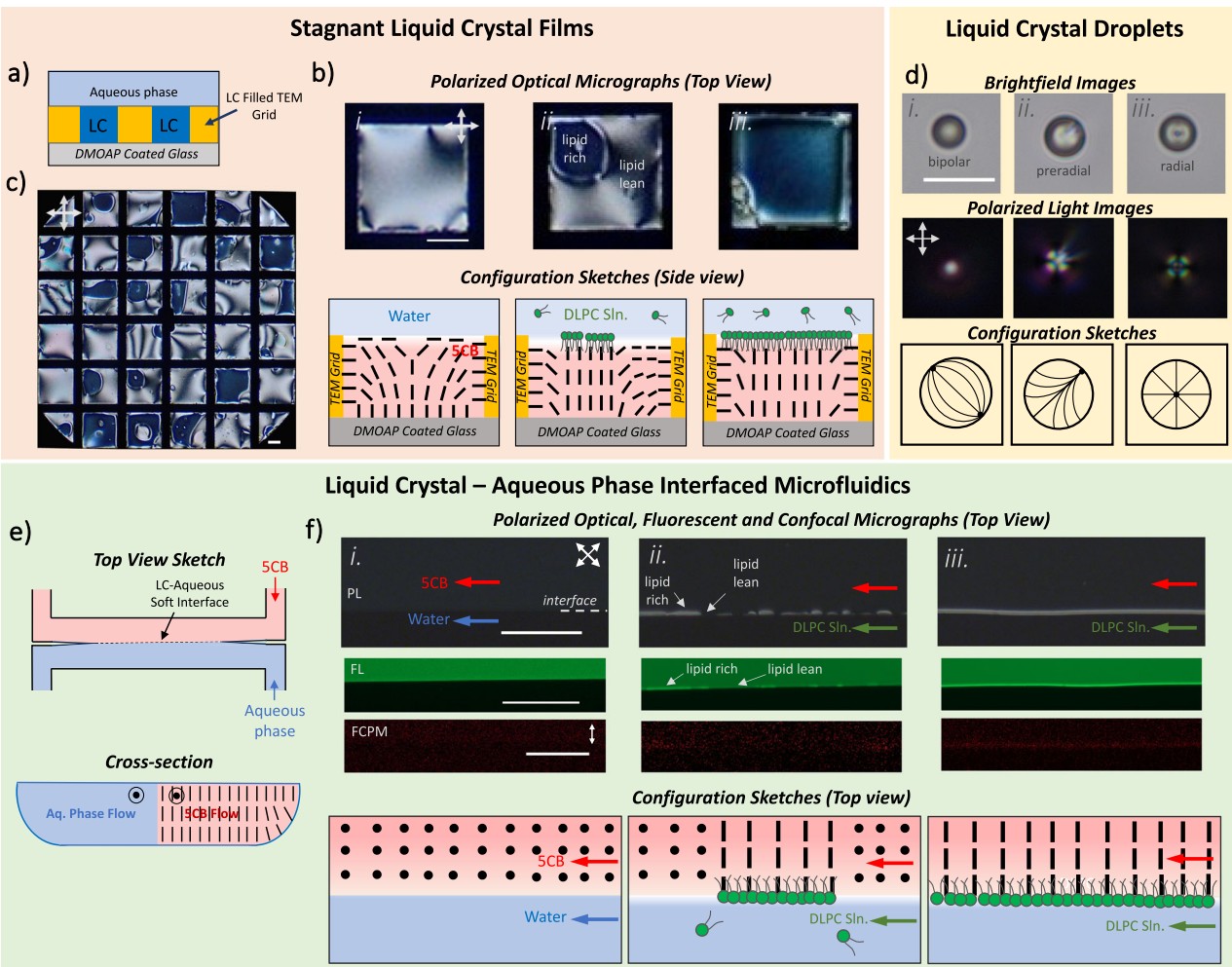

**Fig. 1 | The three experimental systems used in the study. a** Schematic of the system for stagnant LCs supported in TEM grids, **b** representative polarized optical micrographs (top) and cross-sectional configuration sketches (bottom) of the LC-aqueous interfaces with (i) no lipid adsorption, (ii) partial lipid adsorption, and (iii) monolayer of lipids, **c** a representative polarized optical micrograph of an LC supported by TEM grid with partial lipid adsorption. **d** Representative polarized optical micrographs (top), bright field optical micrographs (middle), and configuration sketches (bottom) of (i) bipolar, (ii) preradial, and (iii) radial configurations of 5CB droplets dispersed in PBS. **e** Schematic of the system for microfluidic LC-aqueous interfaces, **f** representative polarized optical micrographs (top), fluorescent micrographs (second row, BODIPY-DLPE), fluorescent confocal micrographs (third row, Nile Red-doped 5CB), and cross-sectional configuration sketches (bottom) of the microfluidic LC-aqueous interfaces with (i) no lipid adsorption, (ii) partial lipid adsorption, and (iii) monolayer of lipids. For the stagnant systems, 5.0 μM DLPC solution (in PBS), and for the flow and droplet systems, 0.5 μM DLPC solution (in PBS) was used. Stagnant systems, flow, and droplet systems were equilibrated at 70, 40, and 80 min, respectively. (Scale bars: 100 μm).

continuously sensing physicochemical changes in the cell membranes, which are important for understanding the diseases related to membrane rigidity, including neurodegenerative diseases, metabolic disorders, and cancer.

## Results and discussion
### The three systems of interest

We used the nematic LC phase of 5CB in both stagnant films, droplets, and microfluidic flow systems to reveal the response characteristics of the LC phases upon fusion of lipid vesicles to the LC-aqueous interfaces. The stagnant film systems were formed by filling 5CB into the wells created by TEM grids supported by a DMOAP-coated glass (Fig. 1a). The DMOAP coating is known to promote homeotropic anchoring, and we confirmed this with the dark appearance of the 5CB in air (also homeotropic) under crossed polarizers. Immersion of the film in pure water resulted in a bright optical appearance of 5CB under crossed polarizers consistent with its planar anchoring at the aqueous interface (Fig. 1b-i). We exchanged half of the aqueous phase with DLPC solution in PBS to maintain a final

concentration of 5.0 μM DLPC. After allowing the fusion of the lipid vesicles and the adsorption of the lipids to the 5CB interface with time (DLPC in this experiment), the dark regions form in the cross-polarized optical images, which are indications of the homeotropic anchoring at aqueous interfaces. This optical appearance was associated with the formation of the monolayer domains of DLPC at the LC-aqueous interfaces, causing homeotropic anchoring, as shown in previous studies[30]. We identify these dark regions as "lipid-rich" domains, and the interface with these phase-separated lipid-rich and lipid-lean domains is named as the "partial coverage" (Fig. 1b-ii). We observed the lipid-rich domains interact with each other and combine, forming a randomly distributed lipid-rich domain at the LC interfaces, as shown in Fig. 1c. With time, more lipids adsorb to the interface, and domains grow and result in full interfacial coverage, resulting in a complete dark appearance under crossed polarizers (Fig. 1b-iii). During the adsorption of lipids to the LC-aqueous interfaces, we observed the signatures of the lipid collision and fusion of vesicles to the LC interfaces, as previously reported[32]. We note that we observe the formation of randomly distributed lipid-rich domains at the LC-aqueous interfaces with the other lipids we

studied; however, the timescales of the formation of the lipid-rich domains were dependent on their type, which is correlated with the melting temperatures (therefore to their phases at the experimental conditions) that we also show below.

For the droplet systems, we formed nematic LC droplets by adding 3 μL of 5CB in 1 mL of 0.5 μM DLPC solution (Fig. 1d). In PBS solutions, LC droplets mostly adopted a bipolar configuration (Fig. 1d-i). With the DLPC adsorption to the droplet interfaces, droplets transformed to mostly the preradial configuration, indicating "partial coverage" (Fig. 1d-ii). With time, DLPC covered the interface of the droplets, forming a homeotropic interfacial anchoring of 5CB, and the droplets adapted mostly radial configurations (Fig. 1d-iii). Such observations were consistent with the past studies[66], which form the basis for our further investigations on the influence of the mechanical properties of the lipid vesicles on their fusion to the interface and, therefore, the LC response characteristics.

For the microfluidic flow systems, 5CB and the aqueous phases flowed simultaneously in a 14 μm-deep microchannel, forming a stable 5CB-aqueous interface (Fig. 1e). The interface formed due to the preferential wetting of the 5CB and the aqueous phases on the DMOAP-coated (hydrophobic) and bare glass/PDMS (hydrophilic) surfaces of the channels, respectively (Fig.1e), stabilized by the Laplace pressure at the interface as explained in our previous studies[64,65]. The dark appearance of the 5CB side under crossed polarizers is apparent in Fig. 1f due to the homeotropic anchoring of 5CB at the DMOAP-coated regions. It was also due to minimal distortion of the LC director with the flow since low inlet pressures (15 mbar) corresponding to low average velocities (~8 μm/s) were used. We also show that the 5CB adopted a planar anchoring at aqueous interfaces, as in the stagnant systems, as supported by the dark appearance of the 5CB-aqueous interfaces under crossed polarizers when imaged from top (Fig. 1f-i). The cross-sectional configuration of the LC director corresponding to this state is shown in Fig. 1e. The dark appearance of the LC-aqueous interface in channels under crossed polarizers is distinct from stagnant systems since LC mesogens are aligned parallel to the aqueous interface but perpendicular to the imaging plane. When the aqueous phase was introduced as a 0.5 μM DLPC solution, we observed the formation of bright domains ("lipid-rich" interface) at the aqueous interfaces (Fig. 1f-ii). The DLPC adsorption to the flowing LC-aqueous interface was assisted with inlet pressures of $P_{aq.} = 5$ mbar and $P_{5CB} = 15$ mbar. These domains were associated with the adsorption of DLPC to the flowing interface, which causes a homeotropic interfacial anchoring of 5CB as sketched in the bottom row of Fig. 1f. We also observed a similar trend in microchannels as in the stagnant systems formed within TEM grids; as more lipids were adsorbed to the interface with time, the interface reached a full coverage of lipids, revealed by the fully bright appearance of the 5CB-aqueous interface (Fig. 1f-iii).

Supporting the homeotropic anchoring of the 5CB at the lipid-rich domains in microchannels, a dark appearance was obtained at the inside of the adsorbed DLPC-rich domains when the arrangements of the analyzers and polarizers were 0° and 90°, respectively (Supplementary Fig. S2a). The sides of the domains appeared bright due to the strained local directors. For the fluorescence microscopy imaging, BODIPY-labeled DHPE was mixed with the DLPC solution to maintain a 1% BODIPY-DHPE. As seen in the insets of Fig. 1f, the fluorescence intensity at the 5CB aqueous interface was increased significantly at the regions where we observed a bright appearance in the crossed polarized images. We utilized fluorescence confocal polarized microscopy (FCPM) to provide further evidence on the presence of the DLPC and alteration of the LC anchoring at the interface with lipid adsorption. We used Nile Red fluorophore mixed with 5CB in a composition of 0.01%. FCPM images taken with and without the presence of adsorbed lipids were indistinguishable at 0° polarization since there was no expected alignment parallel to the direction of excitation polarization (Supplementary Fig. S2b). However, when the images of the same interfaces were collected at 45°, 90°, and 135° polarization of the excitation light (Fig. 1f for 90° images and Supplementary Fig. S2c, d for the rest), we observed higher fluorescence signals at the interface at these polarizer angles supporting the change of the interfacial anchoring of LC from planar to

homeotropic with the adsorption of DLPC. The images collected from the same location without the presence of DLPC did not reveal a measurable intensity difference at the interfaces, supporting the alignment change caused by the presence of DLPC.

## Characterization of LC alignment at the aqueous interface due to the presence of lipids

We characterized the DLPC-rich domains at the LC-aqueous interfaces by using polarized optical micrographs and analyzing the intensity distribution from FCPM images at 45° and 135° polarization. Our first observation was that there were commonly two types of DLPC-rich domains. The first type of domains shown in Fig. 2a-i appeared bright inside under the 45° analyzer and 135° polarizer, while the second domain shown in Fig. 2a-ii (small domain on the right side) showed dark brushes inside. When the arrangements of the analyzers and polarizers were 0° and 90°, a dark appearance was obtained inside the first domain (Fig. 2a-i), and the sides of the domain were bright due to the strained local directors. However, the second DLPC-rich domain (second image in Fig. 2a-ii) appeared mostly bright under 0° analyzer and 90° polarizer. These domains were further analyzed by using FCPM images at 45° and 135°. For the first domain, we found that FCPM images at 45° polarization resulted in a higher intensity at the left side of the domain, whereas imaging at 135° polarization resulted in a higher intensity at the right side of the domain, as given in the intensity plots in the FCPM images of Fig. 2a-i. The trend in the fluorescence intensity was the opposite for the domain in Fig. 2a-ii. Based on the intensity trends, the average configuration of LC mesogens was illustrated in Fig. 2b. The first domain was labeled as "type A", while the other was referred to as "type B". We also observed that, unlike the type A domains (Fig. 2a-i), the type B domains do not form independently; they consistently moved together with type A domains, as shown in in Fig. 2a-ii, suggesting the presence of a defective structure between these domains. Within the samples we investigated, approximately 66% of the identified 80 DLPC-rich domains were "type A" domains, and 22% were "type B" domains, which were always located near the type A domains (Fig. 2c).

## Spatiotemporal characteristics of the adsorption of lipids to the LC-aqueous interfaces

We used polarized optical microscopy to reveal the spatiotemporal response of the 5CB-aqueous interface upon adsorption of DLPC. In addition to the formation of the DLPC-rich domains at the interface, we found that the distribution of the domains was not random as in the stagnant grid system exemplified in Fig. 1c. For representation of this non-uniformity, we report the results by dividing the interface into four distinct regions with the same distance intervals as sketched in Fig. 3a. We observed the formation of the initial DLPC-rich domains towards the exit of the channel ($n = 8$) as shown as "location 4" in Fig. 3a. The images also show that the fractional coverage increased with time as the vesicles fuse to the LC-aqueous interface, leading to the propagation of the front of the interfacial DLPC-rich region towards the inlet of the channel as evidenced by the presence of the DLPC-rich region in "location 3" at $t = 40$ min. Such time scales were significantly different than the anchoring transitions we observed in experiments with singly dispersed surfactants in past experiments, which took less than 1 min[64,65]. We also showed that the formation of the DLPC-rich domains was increased by increasing the DLPC concentration of the aqueous phase, consistent with the increased rate of the fusion of vesicles to the LC-aqueous interface (Fig. 3b). However, the formation of the first DLPC-rich domains close to the exit of the channel was independent of the DLPC concentration in the aqueous phase (Supplementary Fig. S3).

We investigated the formation characteristics of the DLPC-rich domains at the interfaces of stagnant LC systems. We observed a lower rate of formation of the DLPC-rich regions at the stagnant 5CB film interfaces, when compared to the microfluidic channels. The images shown in Fig. 3c were collected from an experiment conducted with a 5.0 μM DLPC equilibrated with the film of 5CB. As shown, even with the more concentrated DLPC vesicle suspensions, the formation of the partial coverage was

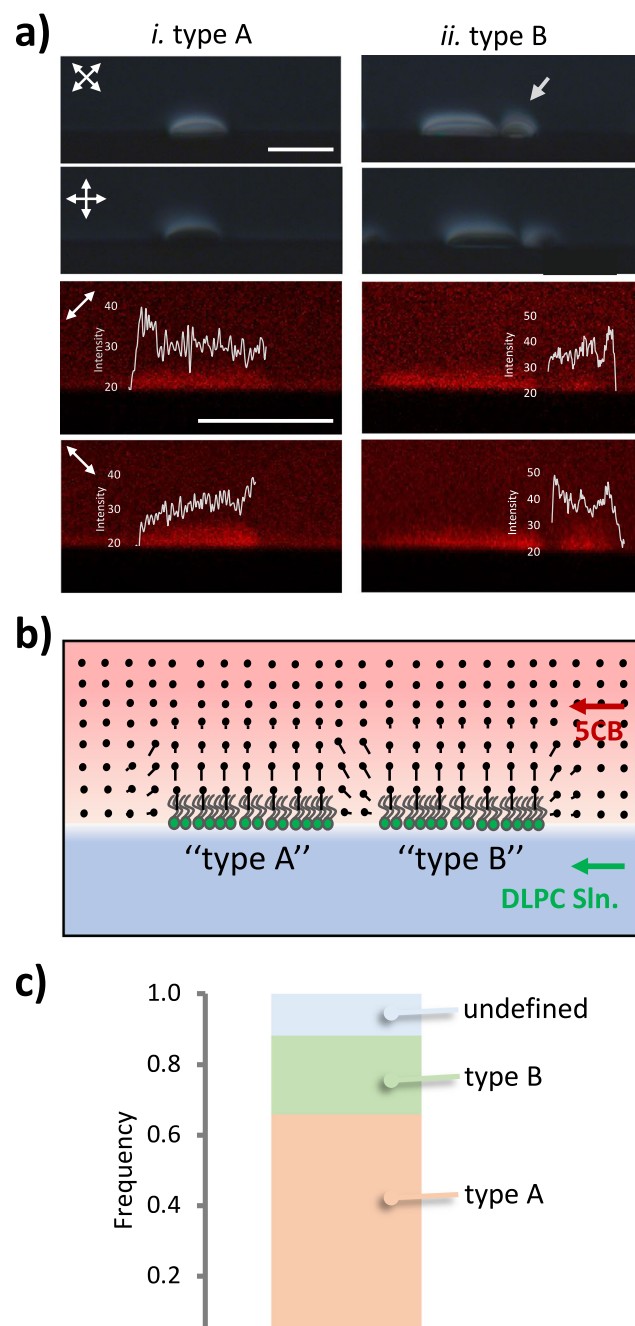

**Fig. 2 | Characterization of LC alignment at the interfaces due to the presence of lipids. a** Representative polarized optical micrograph and FCPM images of DLPC-adsorbed interfaces of microfluidic channels. The directions of the analyzers and polarizers of the polarized optical images are shown in the images with double-sided arrows. The polarization direction of the excitation light of the FCPM images is shown with a single double-sided arrow in the images. Scale bars: 20 μm. **b** Sketch of representative LC configurations in the vicinity of the adsorbed lipid-rich (DLPC) domains. **c** The frequencies of the types of LC configurations are defined in the vicinity of the lipid-rich regions. The data was collected from 80 lipid-rich domains.

maintained in 70 min, which is significantly slower than the experiments conducted with microfluidic systems. In 5CB droplets, we first observed a significant fraction of the droplets to maintain a preradial or bipolar configuration, which exhibit transition into a radial configuration over time, consistent with the formation of a monolayer of DLPC at their interfaces (Fig. 3d, from 0.5 μM DLPC). This rate of configuration change is

comparable to the microfluidic experiments, highlighting the importance of the flow in facilitating the fusion of the lipid vesicles to the LC interfaces. Thus, we concluded that the formation characteristics of the DLPC monolayers at the LC interfaces correlated with the dynamics of the systems we investigated.

We calculated the Peclet number (Pe) as >3000 in microfluidic systems ($Pe = uL/D$, where $L$ is the characteristic length, $u$ is the average flow velocity measured as 135 μm/s, $D$ is the mass diffusion coefficient measured as $1.13 \pm 0.23$ μm$^2$/s for DLPC), indicating the dominant effect of the advective transport mechanism over the diffusive transport. This evidence suggests that the transport of the lipid vesicles to the interface was limited to the fraction that is close to the LC-aqueous interfaces during flow. Thus, the formation of the initial lipid-rich domains towards the exit of the channel highlights the role of the finite fusion rates of the droplets to the LC-aqueous interface. To this end, we performed qualitative COMSOL simulations using the measured dimensions of the channels and the bulk, interfacial velocities, and diffusivities measured experimentally (simulation details explained in Supplementary Information). We found that the formation of the lipid adsorbate concentrates starts from the exit of the channel and propagates towards the inlet with time. Although this result captured the major characteristics of the system, it was *qualitatively* consistent with our main observation since the simulations did not take into account the elastic properties of the "oil" phase (detailed in SI, Supplementary Fig. S4). We found the propagation rate of the interfacial lipid adsorbed layer is strongly dependent on the fusion rate of the lipids at the interface, characterized by a "reaction constant" that designates the fusion of the vesicles to the interface. With this evidence, we next sought to experiment with the flowing LC interfaces by varying the type of lipid, which determines the elastic properties of the formed lipid vesicles[67].

### Adsorption of different types of lipids to the microfluidic soft interfaces

We questioned the response of the LC-aqueous interface against the adsorption of DOPC, Egg SM, and DPPC because the melting temperatures ($T_m$) of the vesicles formed by these lipids differ, as indicated in Table 1. DLPC and DOPC vesicles exhibit liquid-like properties, whereas DPPC and Egg SM vesicles exhibit gel-like properties at room conditions with different elasticities[67–70]. We prepared the lipid vesicle suspensions of 0.5 μM lipids in PBS and characterized the temporal response characteristics of the LCs upon their adsorption.

For DOPC vesicles, the stagnant LC-water interface reached partial coverage within ~70 min (5.0 μM lipids), and the flowing soft interface achieved a similar partial coverage within 40 min, as shown in Fig. 4a (0.5 μM lipids). LC droplets in the DOPC solution (0.5 μM) also showed increasing radial configuration with time, indicating continuing fusion of vesicles to the interface (Fig. 4d). These observations were similar to those in the case of the DLPC adsorption explained above.

In experiments with Egg SM vesicles, we observed no significant adsorption evidenced by the absence of the anchoring transition of the 5CB at the interface for both stagnant and flow systems (Fig. 4b). This lack of fusion of Egg SM vesicles to the interface in channels was reasoned to be the gel phase of the Egg SM vesicles at room temperature due to its ceramide backbone[70]. We note that we confirmed the adsorbed Egg SM monolayers to cause homeotropic anchoring when they were softened with surfactants (DTAB) to aid the delivery of the vesicles to the interface, as done in the literature with DPPC (Supplementary Fig. S5)[30]. 5CB droplet experiments with Egg SM vesicle dispersions also showed no significant fusion of Egg SM vesicles to the 5CB interface, causing a small fraction of pre-radial droplet configurations, as shown in Fig. 4d.

In experiments with DPPC vesicles, which also possess a gel-like state at room temperature, our expectation was the lack of their fusion to the interface. In the stagnant systems, we observed their lack of adsorption within 70 min by using 5.0 μM solution of DPPC (Fig. 4c). However, we observed a clear change in interfacial anchoring of 5CB within 40 min in the microfluidic systems caused by their adsorption (Fig. 4c). This observation

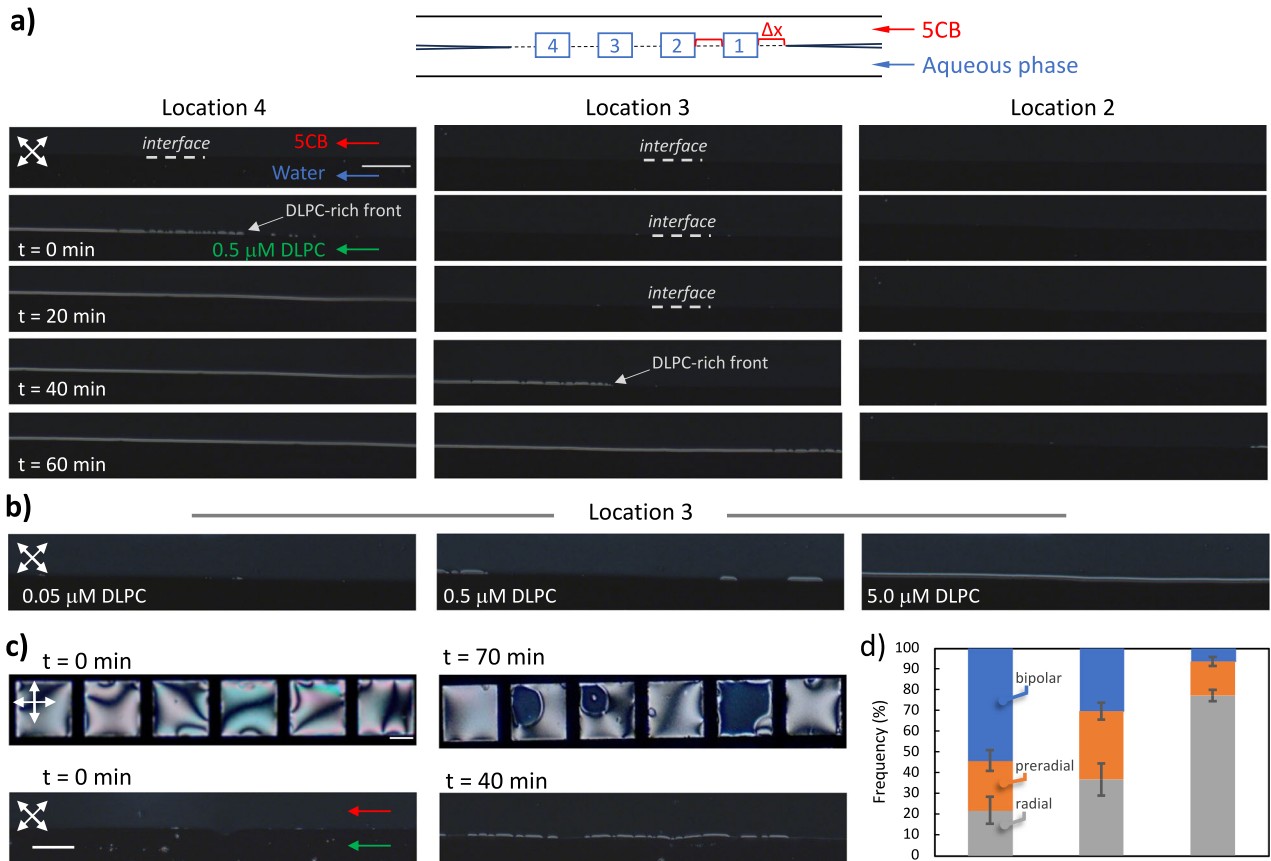

**Fig. 3 | Response of the microfluidic, stagnant, and emulsion systems upon adsorption of the DLPC to the interface.** The microfluidic channel (14 μm) is divided into four locations named 1, 2, 3, and 4 for polarized imaging, where $\Delta x = 1300$ μm. **a** The polarized optical micrographs of the microfluidic system were taken every 10 min for 1 h, where the aqueous phase was introduced as 0.5 μM DLPC ($n = 8$). **b** The polarized optical micrographs of the microfluidic system taken at 40 min from location 3 upon the introduction of the aqueous phase with 0.05, 0.5, and 5.0μM. **c** The polarized optical micrographs of stagnant films and microfluidic systems at the indicated time points and concentrations collected from location 4. **d** Droplet configuration distributions for emulsions equilibrated with 0.5 μM DLPC solution. Scale bars = 100 μm.

**Table 1 | Chemical structures and melting temperatures of phospholipid bilayers**

| Phospholipid | Chemical structure | $T_m$ (°C) |
|---|---|---|
| DLPC | | −2 |
| DOPC | | −17 |
| DPPC | | 41 |
| Egg SM | | 38 |

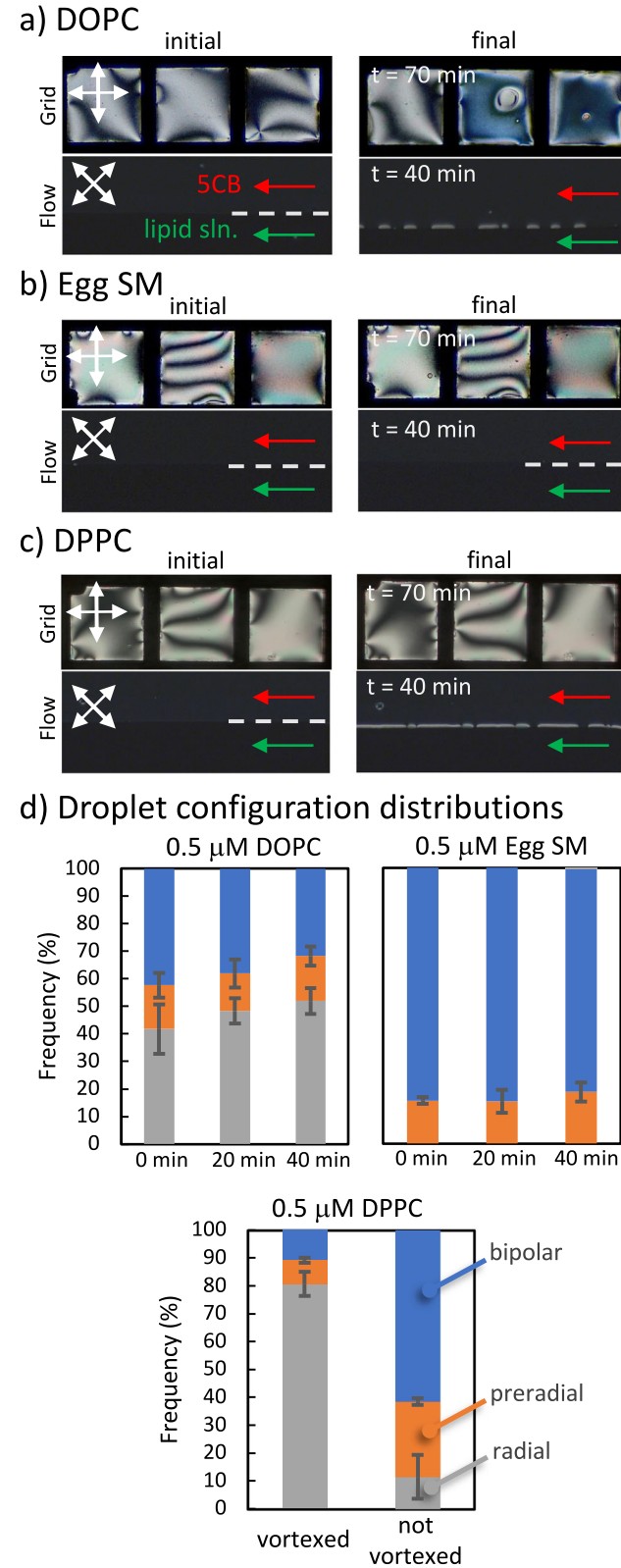

## d) Droplet configuration distributions

**Fig. 4 | Adsorption of different types of lipids into both stagnant and microfluidic LC-aqueous interface.** Polarized optical micrographs were collected under crossed polarize and analyzer 0°−90° for stagnant systems, and 45°−135° for flow systems. Experiments were conducted by equilibrating 5CB with suspensions of **a** DOPC, **b** Egg SM, and **c** DPPC vesicles. The lipid concentrations were 5.0 μM for stagnant and 0.5 μM for microfluidic systems. Final images are collected at 40 min for microfluidic systems and 70 min for stagnant films. Flow was assisted with inlet pressures of $P_{LC}$ = 15 mbar, $P_{Aq.}$ = 5 mbar. **d** Droplet configuration distributions for emulsions equilibrated with 0.5 μM DLPC, Egg SM and DPPC suspensions. Scale bars: 100 μm.

the solution to adopt a majority of bipolar configuration (61.4% ± 9.0%, $n$ = 3) when gently mixed, suggesting a limited adsorption (Fig. 4d). However, for the vigorous shaking case with a vortex, 80.7% ± 4.4% ($n$ = 3) of droplets adopted radial configurations, suggesting a significant fusion of DPPC vesicles to the 5CB-aqueous interface (Fig. 4d). This showed that the rigidity of DPPC vesicles due to their gel phase was able to be overcome by shear. Past studies also showed a finite rigidity[71] of the DPPC monolayers (G" ≈ 5 G')[60]. DPPC gels undergo a structural softening above a critical yield stress, which is 0.2 mN/m, then undergo flow as studied with shear rheology[60]. However, lipid bilayers are typically characterized by their bending rigidity rather than shear surface modulus or viscosity, as bilayers are inherently three-dimensional structures. Therefore, it is more appropriate to correlate our system with rigidity since it consists of the fusion of bilayer to monolayer, and we monitor the fusion of vesicles to the interface by these kinetics. It is shown that DPPC bilayers display greater bending rigidity compared to DOPC bilayers, but are less rigid than Egg SM bilayers[72], which aligned with our observation that the DPPC bilayers can undergo a shear-induced transition out of the gel phase in experiments with microfluidic channels (~200 s$^{-1}$ of shear rate, calculations in Supplementary Information). In contrast, the same shear conditions appear insufficient to induce such a transition in Egg SM bilayers. These observations demonstrated that the microfluidic system can probe the rigidities of the lipid vesicles not only through the temporal changes in the adsorbed amount of the vesicles but also through the controlled shear exerted on the lipid vesicles in the vicinity of the LC-aqueous interfaces.

### Influence of the guest molecules on the fusion characteristics of lipid vesicles at the microfluidic soft interfaces

Observing the influence of the lipid vesicle elasticity on the response characteristics of the stagnant and flowing LC interfaces, we investigated the role of the guest molecules in the lipid bilayers on the response of the microfluidic LC-aqueous interfaces. Dodecyltrimethylammonium bromide (DTAB) is a surfactant known to lower the melting temperatures and increase the fluidity of the lipid bilayers[50]. Past studies showed that DTAB can be used to control the areal density of DLPC by changing the composition of DTAB in DLPC[30]. We monitored the adsorption of DLPC to the flowing 5CB-aqueous interface by mixing 15 μM DTAB, as shown in Fig. 5a ($n$ = 3). We confirmed that a 15 μM DTAB solution was not enough to change the interfacial anchoring of the 5CB from planar when contacted (Fig. 5a-i). When mixed with 0.5 μM DLPC, we observed the formation of the interfacial adsorbed lipids even closer to the channel inlet (location 2) within 20 min, and at location 1 within 40 min. This fusion rate was significantly higher than the fusion of the DLPC vesicles without DTAB (Fig. 5a-ii ($n$ = 3), compared with Fig. 3a). This shows the significant influence of the softening of DLPC vesicles by DTAB. We also demonstrated that increasing DTAB concentration to 150 μM caused a significant increase in the fractional lipid coverage at the interface, forming a full coverage with the introduction of the lipid suspensions ($t$ = 0 min, Supplementary Fig. S6). We conducted the same experiments using Egg SM and DPPC vesicles since they are in the gel phase at room temperature and showed significantly slower adsorption kinetics compared to DLPC. When Egg SM or DPPC was mixed with 15 μM DTAB, we observed a faster adsorption, similar to that evidenced by the significant increase in the fractional coverage of the 5CB-aqueous interface with homeotropic anchoring (Supplementary Fig. S5).

revealed that the shear is likely to overcome the barrier within the gel phase of the DPPC for its fusion, resulting in the adsorption of DPPC to the 5CB-aqueous interface. To understand the influence of shear on the fusion of rigid DPPC vesicles, we conducted LC droplet experiments by both using vortex (~2000 s$^{-1}$ of shear rate, calculations in Supplementary Information) and a gentle handshake (minimal shear). We observed that 5CB droplets in

## a) Softened Vesicles

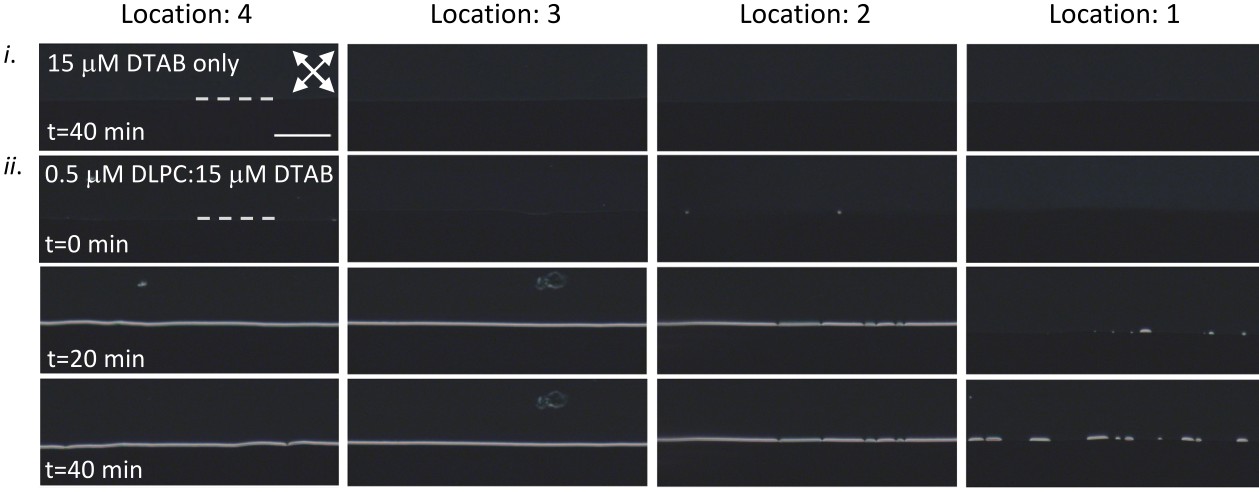

## b) Stiffened Vesicles

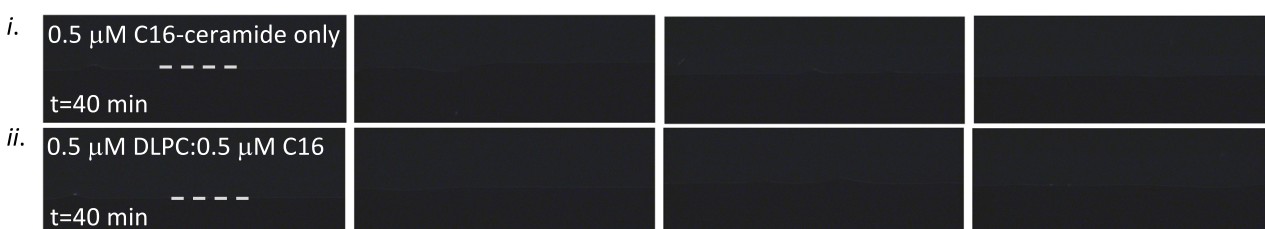

**Fig. 5 | Adsorption of the lipids from softened and stiffened phospholipid vesicles to the flowing LC-aqueous interface.** The polarized optical micrographs shown were taken from the 14 µm-deep microfluidic channels where the aqueous phases were composed of vesicles formed by **a** 0.5 µM DLPC and 15 µM DTAB, and **b** 0.5 µM DLPC and 0.5 µM C16-ceramide. Each line of images was collected from locations 1 to 4 as indicated in the top title. Images collected from experiments with 15 µM DTAB and 0.5 µM C16-ceramide only vesicles are shown in the first row (i) of each figure. Images collected from experiments with 0.5 µM DLPC:15 µM DTAB and 0.5 µM DLPC: 0.5 µM C16-ceramide were collected at the duration after aqueous phase flow was initiated, as indicated in the images. The white double-sided arrows show the orientations of the analyzer and polarizers used in all of the images shown in the figure. The dashed lines in the images indicate the location of the horizontal soft interfaces. Scale bar: 100 µm, common for all images.

Ceramides have high melting temperatures when compared with other lipids ($T_m = 90\,°C$), and the viscoelastic properties of their monolayers showed that ceramide bilayers are solid-like with high shear modulus between 30 and 100 mN/m and high yield stress (100 mN/m)[60]. Moreover, increasing amounts of ceramides resulted in high compression modulus on Egg SM monolayers, mostly due to the effect of condensation[73]. To maintain the rigidifying effect of ceramides on bilayers, we mixed 0.5 µM DLPC with 0.5 µM C16-ceramide[74]. We first confirmed that 0.5 µM C16-ceramide did not significantly fuse at the 5CB-aqueous interface, thus, did not cause an ordering transition of 5CB at the aqueous interface (Fig. 5b-i). We observed that fusion of vesicles did not occur with the mixtures of 0.5 µM DLPC with 0.5 µM C16-ceramide, even at the end of 40 min, as shown in Fig. 5b-ii ($n = 3$). This was consistent with the stiffening of the DLPC vesicles with the introduction of C16-ceramide, making their fusion unfavorable. Also, when we decreased the C16-ceramide amount to 0.16 µM, we observed an increased fractional coverage, indicating fusion of the vesicles to the LC-aqueous interface (Supplementary Fig. S7). These characteristics of DLPC vesicles with 0.16 µM C16-ceramide exhibited slower fusion kinetics when compared with the fusion kinetics of vesicles formed by only DLPC (Fig. 3a), as reflected in both the temporal profile and difference in interfacial lipid coverages. Therefore, we concluded that the response of the interface is strongly dependent on the elastic properties of the vesicles, which influence their fusion rates.

Using the flowing LC-aqueous interfaced microfluidic platform, we sought to investigate the fusion characteristics of the cholesterol-hosted lipid vesicles, as the literature shows its non-universal effect on the elasticity of the host lipid vesicles and its importance in maintaining the structure and function of mammalian cell membranes[20]. We first made observations using vesicles of DLPC doped with cholesterol. We note that the cholesterol did not cause an ordering transition at the interfaces of the LCs, confirmed by the experiments with the microfluidic channels (Supplementary Fig. S8) and droplets, as evident by the bipolar configuration (Fig. 6c). Figure 6a shows the polarized optical micrographs collected after 40 min of contact of the interface with the 0.5 µM DLPC vesicle suspensions with and without cholesterol (see Supplementary Fig. S9 for more detailed temporal data). To compare the fusion rates, we quantified the fractional coverage of the microfluidic 5CB-aqueous interfaces with these mixtures at $t = 40$ min, where we observed a 94% ± 8% ($n = 7$) coverage with DLPC vesicles with no doped cholesterol. With the introduction of cholesterol to the DLPC vesicles, we observed a decreasing fractional coverage at the LC-aqueous interface, as quantified in Fig. 6b. When the cholesterol content was increased to 50%, we observed no evidence of the adsorbed DLPC at the LC-aqueous interfaces ($n = 9$). Above this concentration of cholesterol, addition of cholesterol to the solution resulted in an increasing fractional coverage, reaching 73% ± 28%, 98% ± 3%, and 99% ± 0.9% for 60%, 66% and 75% cholesterol content in 0.5 µM DLPC vesicle suspensions, respectively (Fig. 6b and Supplementary Fig. S10a for temporal data). A similar trend was also observed in the droplets (Fig. 6c). In the case of only 0.5 µM DLPC, radial configuration was observed in 77% ± 3% of the droplets. This percentage decreased to 32% ± 4% upon the addition of 25% cholesterol. The

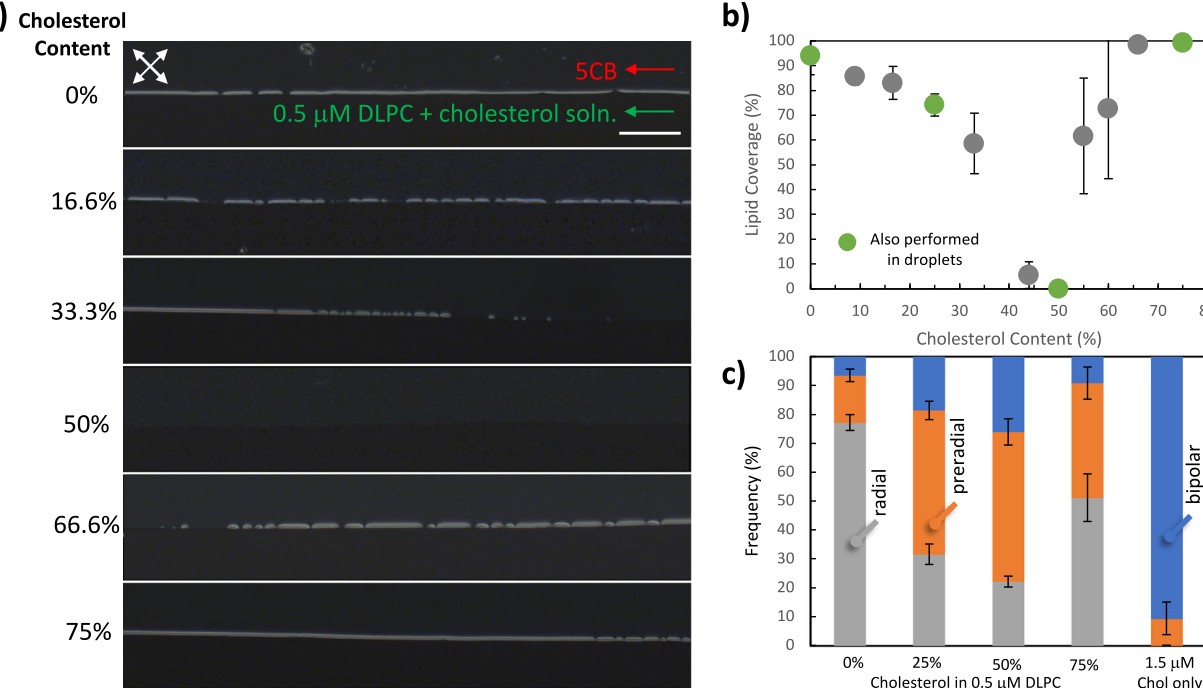

**Fig. 6 | Adsorption of lipids from DLPC-cholesterol mixtures to microfluidic LC-aqueous interface. a** Polarized optical micrographs collected from experiments with contacting 5CB with a constant 0.5 μM DLPC concentration and varying concentrations of cholesterol from 0 to 1.5 μM. Images are shown at $t = 40$ min and at location 4 of the channel. Flow was assisted with inlet pressures of $P_{LC} = 15$ mbar, $P_{Aq.} = 5$ mbar. The images were collected under crossed polarizer and analyzer at 45°−135° as indicated by the two double-sided arrows. **b** Plot of lipid coverage (%) vs. cholesterol content (%) determined from the polarized optical micrographs ($n = 9$). **c** Droplet configuration frequencies with the aqueous phase consisting of a constant 0.5 μM DLPC concentration and varying concentrations of cholesterol from 0 to 1.5 μM ($n = 3$).

lowest occurrence of radial configuration, 22% ± 2%, was observed with 50% cholesterol, which then increased to 51% ± 8% at 75% cholesterol.

The results presented in Fig. 6a-b indicated two major outcomes. First, the mixtures of cholesterol and DLPC cause planar to homeotropic transitions in LCs at aqueous interfaces. Second, combining our observations of the non-monotonous change of the interfacial coverage with the cholesterol concentrations and the influence of the vesicle elasticity on the adsorption characteristics of lipids to the LC-aqueous interfaces explained above, we reasoned that the cholesterol concentration-dependent fusion characteristics of the DLPC vesicles occurred due to the changes in the elastic properties of the DLPC vesicles as a function of the cholesterol concentration. Introduction of cholesterol is known to perturb the lipid bilayer, resulting in changes in the packing of lipids in the bilayer[75]. These changes are highly controversial, as discussed with a range of techniques used in the measurements of the vesicle rigidity[58,76]. The effect of the cholesterol was found to be dependent significantly on the chemical structure of the phospholipids, which was explained in most studies as the saturation of the alkyl chain groups[20,53,54]. Considering the presence of the saturated acyl chains of DLPC, it can be expected to undergo an increase in bending rigidity with the cholesterol content[20]. To understand the effect of cholesterol on DLPC molecules, we measured diffusion coefficients of DLPC with increasing cholesterol amount in the solution (Supplementary Fig. S11a). We found that there is a negligible change in the diffusion coefficient with increasing cholesterol amount, which suggests the difference in the fusion kinetics of DLPC vesicles with cholesterol is related to the changing rigidity of the vesicles. Thus, we concluded that the rigidity of DLPC vesicles increased as the cholesterol content was increased up to 50% based on the adsorption rate we observed. Above this cholesterol loading, vesicles softened that resulting in enhanced fusion rate. These results were consistent with the literature data[20] obtained via NMR measurements, which indicates $K_c$ increases with cholesterol for saturated lipids, but contradicted data using micropipette aspiration[77]. To further analyze this trend, we obtained surface

rheological responses of monolayers formed by pure and cholesterol-doped phospholipids using oscillatory barrier mode Langmuir Trough experiments (Supplementary Fig. S12). The storage (G') and loss (G") moduli exhibit a measurable dependence on lipid composition, type, and cholesterol content in the mixture, indicating pronounced changes in viscoelastic behavior[78]. For DLPC monolayers, Langmuir trough measurements reveal a predominantly viscous, fluid-like behavior as indicated by a higher G" than G' (Supplementary Fig. S13a). Upon introduction of the cholesterol, G' increases and becomes comparable to G", demonstrating a shift in mechanical properties toward a more elastic interfacial response. Moreover, the magnitudes of the complex elastic moduli (Supplementary Fig. S14a) suggested the trend we obtained via our microfluidic and droplet experiments (Fig. 6).

Our observations support the non-universal impact of cholesterol on phospholipid bilayers, given the literature, showing that the LC-aqueous interfaced microfluidics system is sensitive to the changes in chain saturation. However, the lack of literature data is that measurements were completed until 50% cholesterol content, which avoids the effect of the cholesterol-rich phase of lipid vesicles. Our results can be related to the umbrella model and the condensing effect of cholesterol[79,80] on phospholipids, especially after 50% cholesterol amount of the vesicles. In this model, cholesterol perturbs the bilayer[75], and since it has a small hydrophilic head group when compared with the phospholipid molecules, the phospholipid head groups protect cholesterol from the water outside of the bilayer. With increasing cholesterol amounts in the bilayer, cholesterol-cholesterol interactions become unfavorable. After a critical point, cholesterol forms monohydrates. In our results, this critical point was 50 mol% cholesterol amounts, which was observed to be the most rigid composition. Since vesicles are highly rigid in this state, they were not able to fuse into monolayers and adsorbed to the interface, indicated by no visible alignment change of LC mesogens at the interface. After forming monohydrates, cholesterol molecules would not be able to interact with phospholipids.

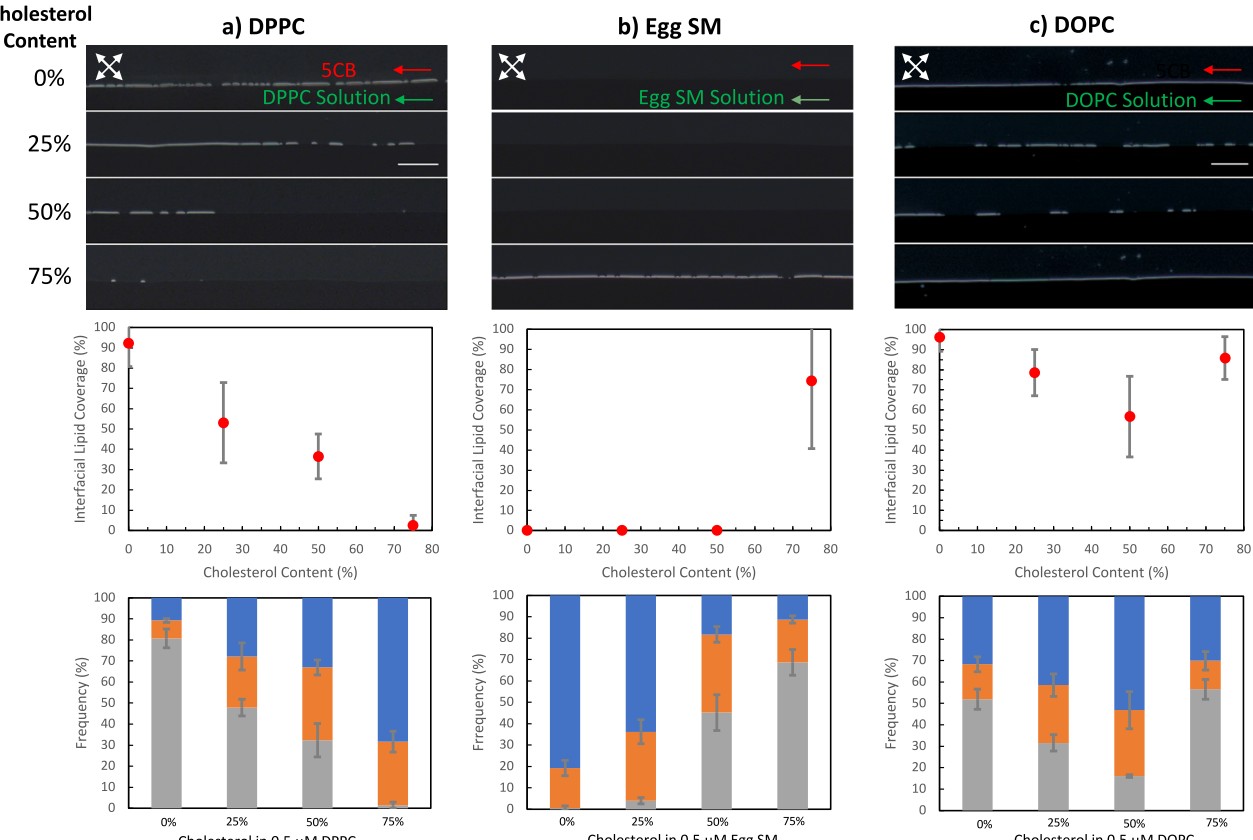

**Fig. 7 | Response of the LC-aqueous interfaces upon adsorption of DOPC, DPPC, and Egg SM from their vesicles with cholesterol as the guest molecule.** The polarized optical micrographs collected at $t = 40$ min and at location 4 are shown in the top panel for the cases where the aqueous phase consists of constant concentrations (0.5 μM) of **a** DPPC, **b** Egg SM, and **c** DOPC, and varying concentrations of cholesterol from 0 to 1.5 μM as indicated. Polarized images were collected under crossed polarizer and analyzer at 45°−135°, respectively. Flow was assisted with inlet pressures of $P_{LC} = 15$ mbar and $P_{Aq.} = 5$ mbar. The middle panels in the figure show the plot of lipid coverage density vs cholesterol content, and the bottom panel shows the configuration frequency (%) of the experiments performed with the droplets incubated in the same concentrations of the lipids and cholesterol. Flow systems: $n = 4$ for DOPC and DPPC; $n = 5$ for Egg SM. For droplet systems: $n = 3$ for each plot. Scale bars are 100 μm, common for all images.

Since vesicles were not rigidified with cholesterol anymore, vesicles were softer than in this state. In our experiments, we observed this effect via a sudden increase in the fractional coverage of the interface with lipids, 61.6% ± 23% from 0%, after 50% cholesterol.

The fusion kinetics of the vesicles formed by cholesterol and lipid mixtures were further investigated by using vesicle suspensions of DOPC, DPPC, and Egg SM in our systems (Fig. 7 and Supplementary Fig. S10 for more details). The same procedures applied to these lipids as in DLPC and similar to the previously explained experimental results, the evidence of the initial adsorption of lipid-cholesterol mixtures to the interface was first observed towards the end of the channels (location 4) as shown in Supplementary Figs. S15 and 17).

As shown in Fig. 7a, we observed a monotonously decreasing fractional coverage trend with the increase in the cholesterol content in DPPC vesicles ($n = 4$). When the droplet response experiments were performed with 5CB droplets (vortexed), we also observed a significant decrease from 80.7% ± 4.4% for 0% cholesterol to 1.4% ± 1.4% for 75% cholesterol in the radial configuration. When compared with the results presented in Fig. 3, which indicated a finite rigidity of the DPPC vesicles as the adsorption was enhanced significantly with the application of shear, the experiments performed in flowing interfaces indicated a significant increase in the rigidity of the DPPC vesicles with the addition of cholesterol. As shown in Supplementary Fig. S11b, the diffusion coefficients of the DPPC vesicles were not influenced significantly or monotonously with the addition of cholesterol. Also, we calculated the Peclet numbers by using measured diffusion

coefficients, and we found that for each cholesterol ratio, it is higher than 2500. Thus, we reasoned that the reduction in the interfacial cholesterol content led to an increase in the rigidity of the vesicles with the addition of cholesterol. This rigidifying effect has been discussed in the literature for DPPC by analyzing interfacial behavior using Langmuir balance techniques, and it is found that DPPC shifts to a lower value of area/molecule monotonously within a range of 0.1 to 0.75 cholesterol fraction due to the condensing effect of cholesterol[81]. Due to its nature, DPPC was hypothesized to act as an "umbrella" strongly with increasing cholesterol perturbation, and thus, we observed a decreasing fractional coverage with increasing cholesterol content in the solution. Furthermore, Langmuir Trough oscillatory measurements reveal that DPPC monolayers exhibit an elastic interfacial response as indicated by a higher G' than G" (Supplementary Fig. S13b), consistently with the literature[78,82]. G" and the magnitude of the complex modulus (Supplementary Fig. S14b) further increases with the cholesterol content of 25%, demonstrating the pronounced stiffening of the monolayer supporting the trend obtained by fusion kinetics.

Egg SM was the most rigid lipid used in our study and showed no evidence of adsorption to the LC-aqueous interface either in droplets of microchannels (Fig. 3). When doped with cholesterol, we did not observe any increase in fractional coverage with the cholesterol amount until 75% cholesterol in microchannels, as shown in Fig. 7b ($n = 4$). At 75% cholesterol, fractional coverage was increased to 74.4% ± 34.4% fractional coverage of the interface. For the Egg SM, we obtained a negligible change in diffusion coefficients with cholesterol (Supplementary Fig. S11c), and Peclet numbers

calculated higher than 2500. These observations and measurements supported the softening of the Egg SM vesicles with the addition of cholesterol content, which is consistent with past studies that reported a decrease in the rigidity of the bilayers of Egg SM with the addition of cholesterol using techniques including flicker spectroscopy and electrodeformation[53]. In droplet systems, the trend was the same with 4% ± 1.4%, 45.2% ± 8.3%, and 68.6% ± 6% for 25%, 50%, and 75% cholesterol, respectively, indicating the decreasing rigidity of Egg SM vesicles with cholesterol. However, radial configuration was observed at 25% and 50% cholesterol in 0.5 µM Egg SM, indicating the adsorption, unlike in microchannels, as shown in Fig. 7b ($n = 4$). This result led us to hypothesize that the observed adsorption of Egg SM in the droplet system was due to enhanced shear. To check this effect, we conducted droplet experiments of 50% cholesterol in 0.5 µM Egg SM by gentle hand-shake and did not observe any significant radial configuration (Supplementary Fig. S18a). Also, by increasing aqueous phase flow rates, these vesicles were able to fuse into the LC-aqueous interface in microchannels (Supplementary Fig. S18b). It is also important to emphasize that the responses measured in the microchannels were more evident than in the droplet experiments, evidencing the influence of the finite rigidity of the cholesterol-doped Egg SM vesicles. Egg SM monolayers exhibited elastic interfacial characteristics with the G' dominating over G", indicating the formation of a rigid monolayer (Supplementary Fig S13c). Upon the cholesterol introduction, the elastic characteristic remained the same with the lower complex elastic modulus value, indicating a decrease in elasticity with the cholesterol of 25%. At the 50% cholesterol content, the monolayer still exhibited a predominantly elastic interfacial characteristic with a relatively higher elastic modulus (Supplementary Fig. S14c).

DOPC showed a similar non-monotonous fractional coverage trend with DLPC but with higher fractional coverage in both microchannels and droplet systems (Fig. 7c). This higher fractional coverage can be explained by the higher fluidity of the DOPC ($n = 4$) when compared with DLPC[67]. For the DOPC, we also obtained a negligible change in diffusion coefficients with cholesterol (Supplementary Fig. S11d), and Peclet numbers calculated higher than 2500 within the range of cholesterol amounts studied. As shown in Supplementary Fig. S13d, DOPC monolayers exhibited a viscoelastic interfacial response with G' and G" being comparable. Upon increasing cholesterol content, both moduli increased, and the enhancement of G' is higher than that of G", indicating a stiffening effect of monolayers (Supplementary Fig. S14d). For the 75% amount of cholesterol, values are still comparable with each other, but with a lower magnitude. The trend of the adsorption was the same for all types of lipids in the stagnant systems, but significantly longer than in flow and droplet systems, as expected (Supplementary Fig. S19).

## Conclusion

We conducted a comprehensive investigation of the fusion characteristics of various biologically relevant synthetic lipids (DLPC, DPPC, DOPC) and biologically-driven lipids (Egg SM) at liquid crystal (LC)–aqueous interfaces, with a focus on how changes in lipid mechanical properties influence their fusion kinetics to the LC interfaces. We successfully established a continuous and responsive platform to provide complementary insights into interfacial lipid organization using both microfluidic, stagnant, and droplet systems. In the microfluidic system, we achieved a stable co-flow of 5CB and an aqueous phase, creating a continuous LC-aqueous interface that enabled real-time observation of lipid adsorption under dynamic conditions. In contrast, stagnant conditions were studied using TEM grid-supported LC films, and droplet experiments involved LC-in-water emulsions. The findings of this study revealed that flow conditions enhance the transport of the phospholipids from the vesicle dispersion to the LC interfaces. We found that altering the rigidity of the lipid vesicles by using surfactants and ceramides resulted in significant changes in their fusion kinetics at the interface. We also explored the effect of cholesterol on lipid bilayer rigidity and its subsequent impact on interfacial characteristics. Interestingly, the influence of cholesterol was found to be non-universal and

lipid-specific. For DLPC and DOPC, cholesterol increased bilayer rigidity up to ~50 mol% DPPC, a lipid in the gel phase at room temperature, exhibited a more gradual, monotonic increase in bilayer rigidity with cholesterol content. In contrast, Egg SM showed an opposite trend, where cholesterol resulted in a decreased rigidity beyond a threshold. The observed fusion trend aligned with the umbrella model and condensing effect on cholesterol in the lipid bilayers[80]. Complementary Langmuir Trough oscillatory measurements revealed lipid-specific and cholesterol-dependent monolayer elasticity that qualitatively supported the trends observed in vesicle rigidity and fusion kinetics. Although the measurements do not directly provide quantification of bilayer rigidity, they provide independent mechanical insights into cholesterol-induced alterations in the mechanical properties of the monolayers. These findings provide new insights into changing membrane mechanics with cholesterol. By combining the recent advances in the literature[64,65], we demonstrated that flowing LC-aqueous interfaces provide highly sensitive platforms for detecting changes in the properties of the lipid vesicles. The microfluidic approach enabled real-time and continuous monitoring of the lipid vesicle fusion to the soft interfaces formed by thermotropic LC and aqueous phases. This system provides new insights for understanding biological membranes in dynamic environments. Recently, LC-aqueous interfaces have emerged as optical biosensing platforms where biomolecular interactions, such as analyte binding and enzymatic reactions, induce a change in the optical appearance of the interface. This system could find a niche in these studies by aiming to investigate intrinsic mechanical properties of membranes rather than external biochemical species and interactions. Considering the biological relevance of the lipid vesicles used and their complexes formed with the guest molecules, integrating this platform with advanced techniques may enhance the study of membrane biomechanics and their role in disease progression. The quantification of mechanical properties of membranes, especially bending rigidity, can be recognized as experimentally challenging due to the sensitivity of lipids and methodological limitations[76]. Our system provides the potential to overcome these challenges by offering insights about the fusion characteristics of vesicles to a soft flowing interface, combined with interfacial analysis. Potential technologies derived based on the demonstrations we show herein may act as a transformative path towards point-of-care or marker-based diagnostic systems for critical or rare diseases, which include neurodegenerative diseases, metabolic disorders, and cancer. In particular, extracellular vesicles have already been used in cancer-related clinical studies as biocargos and biomarkers[13], and their integration with microfluidic technologies enabled the high-throughput and minimally invasive diagnostics[83]. Our study contributes to a platform that can be translated to extracellular research and further enhances detection sensitivity.

## Methods
### Materials

The room temperature nematic liquid crystal 4-cyano-4'-pentylbiphenyl (5CB) was purchased from HCCH Jiangsu Hecheng Chemical Materials Co., Ltd. (Nanjing, China). Glass slides were obtained from Marienfeld GmbH (Lauda-Königshofen, Germany). HPLC-grade acetone, 2-propanol, trichloro(octadecyl)silane (OTS), dimethyloctadecyl[3-(trimethoxysilyl)propyl]ammonium chloride (DMOAP), Nile Red dye, cholesterol (powder), dodecyltrimethylammonium bromide (DTAB), and phosphate-buffered saline (PBS) were purchased from Sigma-Aldrich (St. Louis, MO, USA). 1,2-dilauroyl-sn-glycero-3-phosphocholine (DLPC), 1,2-dipalmitoyl-sn-glycero-3-phosphocholine (DPPC), 1,2-dioleoyl-sn-glycero-3-phosphocholine (DOPC), egg sphingomyelin (Egg SM, Chicken) and N-(4,4-difluoro-5,7-dimethyl-4-bora-3a,4a-diaza-s-indacene-3-propionyl)-1,2-dihexadecanoyl-sn-glycero-3-phosphoethanolamine (BODIPY-DHPE) were purchased from Avanti Polar Lipids. N-((2S,3 R,E)-1,3-Dihydroxyoctadec-4-en-2-yl) palmitamide (C16-Ceramide) was purchased from BLD Pharma (China). Polydimethylsiloxane (PDMS, Sylgard 184 elastomer kit) was purchased from Dow Europe GmbH (Wiesbaden, Germany). The negative photoresist AZ P4620, AZ EBR solvent, AZ 400 K

developer 1:4, and Buffered Oxide Etchant 7:1 (BOE) were purchased from Microchemicals GmbH (Ulm, Germany).

## Fabrication of the microfluidic channels

The sketch of the channel used in the lithography process is shown in Supplementary Fig. S1a, which is composed of two inlets and two outlet ports that combine at the main channel that facilitates the co-flow of the nematic LC with an aqueous phase. The microfabrication procedure of this channel is as follows (Supplementary Fig. S1b). The glass slides were sonicated for 15 min in acetone and 2-isopropanol, respectively, for initial cleaning. Then, glass slides were sonicated in aqueous DMOAP (1% v/v) solution for 10 min to achieve the hydrophobicity necessary for the photoresist adhesion. The photoresist solution was prepared by mixing the photoresist AZ P4620 and EBR solvent in a 1:1 volume ratio. The surface of the DMOAP-coated glasses was coated with a 1:1 photoresist solution by using a Polos spin coater (SPS-Europe B.V., Putten, The Netherlands). The spin speed was set at 1000 rpm for 50 s. Photoresist-coated glass was soft-baked at 110 °C for 50 s and cooled down to room temperature at ambient conditions for approximately 10 min. Polos Nanowriter (SPS-Europe B.V., Putten, The Netherlands) maskless photolithography system was used with an exposure dose of 160 mJ/cm$^2$ laser with a 405 nm wavelength for the channel writing. The surfaces were then treated with AZ 400 K developer solution for 80 s. The developed surfaces were hard-baked for 90 min at 110 °C. After equilibrating at room temperature, wet etching was performed by using a 7:1 BOE solution. Glasses were immersed in a PTFE beaker containing the BOE solution and sonicated for 20 min. The temperature was maintained at room temperature for consistency during the etching process. After etching, patterned glasses were cleaned with fresh distilled deionized water and then rinsed in a jar containing acetone for stripping. The stripping was done in an ultrasonic bath for 10 min. The channel-etched glasses were then rinsed with copious amounts of acetone, water, and 2-propanol, respectively, and dried with a nitrogen stream. We measured the depth of the resulting channels to be 14 μm by using fluorescence confocal polarized imaging.

PDMS used to cover the top layer of the channel-etched glasses was prepared by mixing the elastomer base and curing agent in a 10:1 mass ratio by mixing them homogeneously. The mixture was left for 30 min in a vacuum chamber to remove the bubbles formed during mixing. The prepared mixture was poured into the mold containing OTS-coated flat glass slides (OTS was deposited on glass slides using vacuum deposition for 30 min). The mixtures were cured at 60 °C for 2 h in the oven. After curing, the PDMS was left to cool down to room temperature. A piece that corresponds to the etched microfluidic channel on the glass slide was cut, and four ports were punched in the region corresponding to the inlets and the outlets on the glass slide by using a 1.5 mm biopsy punch. The glass-PDMS bonding was performed by oxygen plasma treatment, Diener Electronics (Ebhausen, Germany). After exposure to oxygen plasma for 20 s, PDMS and the channel-etched glass slides were bonded. For the flow experiments, 1.5 mm PTFE tubing was connected to the inlet ports of the channels. The liquid inlet pressures were controlled by an Elveflow OB1 MK3+ (Paris, France) microfluidic flow control system.

To achieve the stable two-phase flow, DMOAP (from a 1% volume solution in water) was coated to approximately one-half of the channel in its longitudinal direction using flow focusing using 1% DMOAP solution and deionized water co-flow system. The inlet pressure of the DMOAP solution was set to 190 mbar, and 200 mbar was set for water. The simultaneous flows were ensured for 30 min. The channel was initially filled with water flow, and DMOAP solution was introduced with a 3 min ramp. After DMOAP functionalization, the tubing was gently removed. Water flow is ensured for at least 15 min to rinse the surfaces from residual DMOAP solution. While the water was flowing, 5CB tubing was introduced to the side coated with DMOAP. Simultaneous flow creating an LC-aqueous interface was maintained throughout the microfluidic channel. All the flow experiments were conducted at room conditions.

## Preparation of stagnant systems

To create homeotropic anchoring, glass slides were coated with DMOAP following the same procedure explained in the previous section and then cut into small pieces. 25 μm-thick gold-coated TEM grids were placed on DMOAP-coated glass and filled with 5CB. Glass with TEM grid was immersed gently in a cuvette filled with aqueous PBS solution of 2 mL. Then, half of the aqueous phase was exchanged with a lipid solution to achieve the desired concentration indicated in the experiments.

## Preparation of vesicle suspensions

To prepare a lipid solution for the experiments, 0.5 μL of DLPC (25 mg/mL in chloroform) was taken into the glass vial. The gentle flow of the nitrogen stream was used for evaporating the chloroform. After further evaporation of the residual chloroform under vacuum (<50 mbar) for 5 min at 60 °C, 4 mL PBS solution was added to achieve a 5.0 μM DLPC solution and vortexed at 3000 rpm for 5 min to disperse vesicles in suspension. This solution was diluted with a PBS solution to the desired concentration to be used in experiments. The same procedure was applied to other lipids; however, vortex mixing was performed at room temperature for DLPC and DOPC, whereas it was performed at 50 °C for DPPC and Egg SM. Cholesterol (in powder form) solution was prepared with chloroform to achieve a 25 mg/mL solution. 0.6 μL from this solution was taken into a glass vial, and the same procedure was applied with the phospholipid solution preparation. Phospholipids and cholesterol solutions were mixed in desired ratios. To obtain a 0.3 mM DTAB solution, 4.6 mg of DTAB (in powder form) was dissolved in 5 mL of PBS. The DTAB solution was diluted into the desired amount and mixed with phospholipids in the desired ratios. 1 mg C16-ceramide (in powder form) was dissolved in 1 mL of chloroform to obtain a batch solution of C16-ceramide. 11 μL was taken from the batch to the glass vial, and the same procedure was applied with the phospholipid solution preparation. Phospholipids and C16-ceramide solutions were mixed in desired ratios. The diffusion coefficients and average sizes are reported in Fig. S11 as mean ± standard deviation of the three independent measurements. The prepared solutions were fed into the microfluidic channels through a PTFE tubing from the hydrophilic side of the microfluidic channel. Size measurements of these solutions were completed by ZetaSizer Ultra (Malvern Instruments Ltd., USA).

## Microscopy

The microfluidic system was imaged using a Zeiss LSM 900 Fluorescence Confocal Polarizing Microscope (Jena, Germany) equipped with a linear polarizer and an analyzer for confocal and optical imaging with polarized light. Crossed polarizers and analyzers were used at 45° and 135° or 0° and 90° angles in polarized optical microscopy to determine the orientation of the LC local director. Microfluidic channels were located horizontally from the top view for all experiments, and the flow was from right to left as shown in the images. In fluorescence imaging, BODIPY (1 v%) ($\lambda_{ex}$ = 501 nm and $\lambda_{em}$ = 512 nm) labeled DHPE was mixed with the lipids to track their presence. In confocal imaging, Nile Red ($\lambda_{ex}$ = 559 nm and $\lambda_{em}$ = 635 nm) was mixed with 5CB with 0.01 wt% %[84]. The excitation of the laser source was 561 nm wavelength, and the pinhole was 78 μm in FCPM imaging. Images were collected at 4% intensity and 750 V master gain using a 40x objective. For the imaging of the stagnant systems, an Olympus BX53 (Tokyo, Japan) equipped with crossed polarizers was used. Crossed polarizers and analyzers were used at 0° and 90° angles.

## Characterization of the phospholipid monolayers at the water–air interface

Langmuir monolayer experiments were performed using a Langmuir Trough system (KN 2001, Biolin Scientific, Finland) equipped with a platinum Wilhelmy plate (Espoo, Finland) with a wetted length of 39.24 mm and barriers. Prior to each experiment, the trough and barriers were cleaned with ethanol by brushing and rinsed with ultrapure water. The aqueous subphase was filled into the system, and the interfaces were cleaned thoroughly.

Lipid or lipid-cholesterol mixtures, which were prepared in chloroform with a constant amount of lipid as 1 mg/mL, were carefully spread dropwise into the air-water interface using a glass microsyringe in an amount corresponding to the desired molecule area. After spreading, the monolayer was allowed to equilibrate for 15 min to ensure the chloroform evaporation. Following the equilibration, the monolayer was compressed at a constant barrier speed of 10 mm/min. The surface pressure was recorded as a function of mean molecular area to obtain isotherms. For the surface rheological properties, the oscillating barrier method was used. The target surface pressures were selected as 27.5, 22.5, 32.5, and 30 mN/m for DLPC, DOPC, DPPC, and Egg SM, respectively. After compressing the monolayer to the target surface pressure, sinusoidal area oscillations were applied at a frequency of 30 mHz. All data were analyzed using the KSV NIMA LB software (Biolin Scientific).

## Statistical analysis

The sample sizes ($n$) and error bars are defined in the text when needed. The data were reported as mean ± standard deviation.

## Data availability

All raw data supporting the findings of this study are openly available at https://doi.org/10.5281/zenodo.18976028.

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

## Acknowledgements

Financial support from the European Research Council under Starting Grant, LCFlow (grant agreement no. 101039294; awardee, Emre Bukusoglu) is gratefully acknowledged. This work was partially supported by the Research Fund of the Middle East Technical University. Project Number: 11628. The authors thank Dr. Burak Akdeniz for their help in developing the COMSOL simulations.

## Author contributions

C.D. conducted experiments and characterizations. E.B. supervised the research. All authors contributed to data interpretation, discussions, and manuscript preparation.

## Competing interests

The authors declare no competing interests.
