## [Transparent Peer Review file · Communications Materials]

Soft-Interfaced Liquid Crystal Microfluidics Can Probe the Rigidity of Lipid Vesicles

Corresponding Author: Professor Emre Bukusoglu

Version 0:

Decision Letter:

**** Please ensure you delete the link to your author homepage in this email if you wish to forward it to your coauthors ****

Dear Dr Bukusoglu,

Thank you for submitting your manuscript, "Soft-Interfaced Liquid Crystal Microfluidics Can Probe the Rigidity of Lipid Vesicles", to Communications Materials. It has now been seen by 2 referees, whose comments are appended below. You will see that while they find your work of potential interest, they raise substantial concerns that must be addressed. In light of these comments, we cannot accept the manuscript for publication but are interested in considering a revised version that addresses these serious concerns.

To allow us to move forward with your work, we also ask that you edit your manuscript according to the attached table. **Please read this document carefully as we will be unable to further assess your revised paper until these important points are addressed.**

Please outline all revisions made in the right-hand column and return the completed table with your updated manuscript files as a Related Manuscript file.

When resubmitting, please also include:

- A point-by-point response to the reviewers' comments. If you are unable to address specific reviewer requests or find any points invalid, please explain
- A clean version of your revised manuscript with no mark-ups
- A marked-up version of your paper with all changes highlighted in a different colour

Please use the link below to submit your revised files:

Link Redacted

**** This url links to your confidential home page and associated information about manuscripts you may have submitted or that you are reviewing for us. If you wish to forward this email to co-authors, please delete the link to your homepage first ****

We hope to receive your revised paper within twelve weeks, but we understand that revisions may take longer. Please let us know if you find that the revision process will take substantially more time.

We are committed to providing a fair and constructive peer-review process. Please do not hesitate to contact me if you have any questions or would like to discuss these revisions further. We look forward to seeing the revised manuscript and thank you for the opportunity to review your work.

Best regards,

John Plummer, PhD
Chief Editor
orcid.org/0000-0003-4824-8497
Communications Materials

Reviewers' comments:

Reviewer #1 (Remarks to the Author):

The authors in the present article study the mechanical properties of lipid vesicles.

The manuscript is unclear and confusingly written. It is difficult for the reader to understand the logical flowchart of the experiments and therefore their purpose. According to the authors, the purpose is to understand the role of cholesterol on the mechanical properties of phospholipid vesicles.

Reading the experimental section, it becomes apparent that the authors do not use vesicles but an undefined phospholipid suspension. Vesicles are prepared from a phospholipid film in a test tube obtained by the evaporation of a solution containing the lipid and an organic solvent. Then, one proceeds with hydration using the buffer, followed by vortexing at a temperature higher than the gel-to-liquid crystal transition temperature of the lipid used. The vortexing step is the most important because it provides the necessary energy for bilayer formation. Systems containing cholesterol must be prepared in the same way by adding the programmed amount of cholesterol to the organic phase containing the phospholipid. The manuscript in its present form is not suitable for publication.

Reviewer #2 (Remarks to the Author):

The paper deals with an interesting and potentially important problem in the scarcely studied “no man’s land” between thermotropic liquid crystals (widely used, e.g., in electrooptical displays) and phospholipid membranes (which are among fundamental structures in cell biology). The lateral organization of the water-dispersed phospholipids (which is, in fact, liquid crystalline) at the interface between thermotropic liquid crystal and aqueous medium induces interface-driven orientational transitions of the thermotropic LC, in this particular case, the nematic 5CB. These transitions were studied in both “stagnant” films (the meaning is clear, but it is a rather strange term!), droplets, and microfluidic flow systems to record the response characteristics of the LC phases of 5CB upon fusion of lipid vesicles at the interface.

The approach used in this paper is closely related to a series of works where the interface between 5CB and aqueous dispersions of various organic systems, mainly of surfactant nature, was used as an analytical tool providing qualitative (or even quantitative) information on the composition of the aqueous dispersion.

The general level of the paper is rather high, the results obtained for the microfluidic flow mode seem to be sufficiently new, and the narration is thorough and comprehensive.

However, certain remarks should be made.

1. The authors use several types of phospholipid systems (saturated, as DPPC or DMPC, and non-saturated, as DOPC or SOPC. The authors speak of “bending rigidity” K_c as an important parameter affecting the observed properties. Probably there should be some correlation between bending rigidity and phase transition points on DSC thermograms of hydrated phospholipids (which are clearly different for these phospholipids). This aspect should be analyzed in the revised version. Special attention should be paid to the systems phospholipid + cholesterol, for which this question is especially interesting.
2. The abstract should be re-written with substantial changes. In the present form, the scientific novelty of the paper remains obscure – is it the microfluidics as an addition to other method of studies of phospholipid-5CB interactions, or the main sense is in mechanical properties (rigidity) of different phospholipids?
3. The general idea seems to have grown from the papers of Abbot e.a. published in 2005 and even earlier. The progress in this topic during recent 20 years (which is undeniable) should be clearly described.
4. A similar approach has been widely used for another problem – absorption of organics from water dispersions on oriented 5CB structures as means for optical sensing in biomedical applications (there are plenty of papers using similar methods, but without any mentioning of phospholipids). Again, the question – the place of this work among multiple analogues should be clearly outlined.

Provided the above questions and remarks are accounted for in the revised text, the paper could be considered for publication.

** See Nature Research’s author and referees’ website at www.nature.com/authors for information about policies, services and author benefits

Version 1:

Decision Letter:

**** Please ensure you delete the link to your author homepage in this email if you wish to forward it to your coauthors ****

Dear Professor Bukusoglu,

Thank you again for submitting your revised manuscript "Soft-Interfaced Liquid Crystal Microfluidics Can Probe the Rigidity of Lipid Vesicles" to Communications Materials. We have now received reports from the 2 reviewers and, based on their comments, we have decided to invite a revision of your work. You will see that Reviewer 1 still finds the methods unclear.

When resubmitting, please also include:

- A point-by-point response to the reviewers' comments. If you are unable to address specific reviewer requests or find any points invalid, please explain
- A clean version of your revised manuscript with no mark-ups
- A marked-up version of your paper with all changes highlighted in a different colour

Please use the link below to submit your revised files:

Link Redacted

**** This url links to your confidential home page and associated information about manuscripts you may have submitted or that you are reviewing for us. If you wish to forward this email to co-authors, please delete the link to your homepage first ****

We hope to receive your revised paper within six weeks, but we understand that the revisions may take longer. Please let us know if you find that the revision process will take substantially more time.

We are committed to providing a fair and constructive peer-review process. Please do not hesitate to contact me if you have any questions or would like to discuss these revisions further. We look forward to seeing the revised manuscript and thank you for the opportunity to review your work.

Best regards,

John Plummer, PhD
Chief Editor
orcid.org/0000-0003-4824-8497
Communications Materials

Reviewers' comments:

Reviewer #1 (Remarks to the Author):

In their rebuttal letter to the Editor, the authors state the following: "We would like to point out that our vesicle preparation procedure indeed follows the established film hydration method as outlined by the reviewer and the literature. We first evaporate the chloroform containing the lipid, then hydrate it with buffer (PBS), and vortexing is performed to supply the necessary energy to form a bilayer."

However, in both the marked and unmarked versions of the main manuscript, the authors write: "To prepare a lipid solution for the experiments, 0.5 μ L of DLPC (25 mg/mL in chloroform) was taken into the glass vial. The gentle flow of the nitrogen stream was used for evaporating the chloroform. 4 mL PBS solution was added to achieve a 5.0 μ M DLPC solution. This solution was diluted with a PBS solution to the desired concentration used in experiments. The same procedure was applied to other lipids."

Based on this text, there is no mention of vortexing, the preparation temperature, or the use of a vacuum pump for chloroform removal. I cannot verify whether the authors' claims in the letter to the Editor are accurate. As it stands, the experimental description indicates that the prepared sample does not consist of vesicles, but rather an undefined phospholipid suspension.

Furthermore, the authors should be aware that even with vortexing, one typically obtains multilamellar vesicles (MLVs), which have an "onion-like" structure and a large radius of curvature that significantly influences vesicle fusion phenomena. While small vesicles have a strong propensity to fuse, large ones do not. Therefore, reporting the size of the vesicles used is crucial.

I would like to remind the authors that an accurate description of sample preparation is fundamental to the integrity of a

manuscript. An experiment must be reproducible in any laboratory; however, without a detailed description of the experimental conditions, replication is impossible.

Reviewer #2 (Remarks to the Author):

I think that the authors have given satisfactory answers to my questions and remarks. The revised text has been substantially improved, and now I think that the paper should be accepted for publication.

** See Nature Research's author and referees' website at www.nature.com/authors for information about policies, services and author benefits

Version 2:

Decision Letter:

** Please ensure you delete the link to your author homepage in this email if you wish to forward it to your coauthors **

Dear Professor Bukusoglu,

Thank you once again for submitting your manuscript, "Soft-Interfaced Liquid Crystal Microfluidics Can Probe the Rigidity of Lipid Vesicles," to Communications Materials. Your manuscript has been seen again by the referees, whose comments are appended below. I am happy to say that the concerns of our reviewers have now been addressed, and that we only require some minor amendments before we can accept your paper.

Our remaining requests are:

- Please upload all the main figure files as they were uploaded before.
- Please provide the supplementary information file in PDF format.

This will be the final revision of your manuscript. We ask that you carefully review all files associated with your paper and follow the link below to upload the final version of all files, including display items and supplementary material. Please ensure that these files are clean, without any markups or comments, as they will be sent for publication.

Link Redacted

We hope to receive this updated version of your paper within 1 week, but please let us know if you find that you need more time.

We hope to receive this updated version of your paper **within 1-week**, but please let us know if you find that you need more time.

Best regards,
John Plummer, PhD
Chief Editor
orcid.org/0000-0003-4824-8497
Communications Materials

Reviewers' comments:

Reviewer #1 (Remarks to the Author):

The authors have satisfactorily addressed all the issues I raised. In its current form, the manuscript is suitable for publication.

Version 3:

Decision Letter:

Dear Professor Bukusoglu,

We are delighted to accept your manuscript titled "Soft-Interfaced Liquid Crystal Microfluidics Can Probe the Rigidity of Lipid Vesicles" for publication in Communications Materials. Thank you for choosing to publish your interesting work with us.

Acceptance of your manuscript is conditional on all authors' agreement with [our publication policies](https://www.nature.com/commsenv/editorial-policies). In particular, your manuscript must not be published elsewhere and there must be no announcement of the work in the media until the publication date.

Please carefully read the information below for information on what to expect from the next steps of the publishing process:

Article in Press:

Please note that in advance of your paper being published we will host an early access version, known as an 'Article in Press,' on our journal website. For more information on this initiative please see our [author guidelines](https://support.springernature.com/en/support/solutions/articles/6000281821-what-is-an-article-in-press-).

Publication as an [Article in Press](https://support.springernature.com/en/support/solutions/articles/6000281821-what-is-an-article-in-press-) is typically within 1-2 weeks after we have received your corrected proofs and publishing agreement. Subsequently, we will aim to publish the Version of Record in a timely manner. Please note there will be no further correspondence about your publication date.

When your article is published as the Version of Record, you will receive a notification email. **If you are planning an embargoed press release or require a specific publication date, please complete our [scheduling requests form](https://forms.office.com/e/ed7NBDd08u), or contact commsproduction@springernature.com, as soon as possible after acceptance and we will endeavour to accommodate your request.**

For further information on the journey of your article from acceptance to publication, please see our [Author FAQs](https://www.nature.com/documents/Author_FAQs.pdf).

Publishing Agreements and Fees:

In about one week, you will receive an email with a link to complete the appropriate grant of rights necessary for publishing your paper and – if applicable – to provide payment information for your article-processing charge (APC), either via credit card or by requesting an invoice.

If needed, our Author Services team will be in touch regarding any additional information that may be required.

In order to avoid any delays, please ensure that you have emails from Springer Nature whitelisted in your mail system.

Proofs:

At this point we will also edit your manuscript to ensure that it conforms with our house style. Once you have completed the publication agreement and arranged payment, you will receive a separate email with a link to an online eProof for you to review. Please read your proof with great care to ensure that no changes have been introduced which have inadvertently altered the meaning of your paper. We suggest that you discuss the proof with your co-authors, but please ensure that only one author communicates with us and that only one set of corrections is returned via the online correction in the eProof.

The corresponding (or nominated) author is responsible on behalf of all co-authors for the accuracy of all content, including spelling of names and current affiliations.

To ensure prompt publication, your proofs should be returned within two working days. If there is any period within the next four weeks in which you won't be available, please nominate a co-author with whom we can correspond and send their contact information to us via email at commsproduction@springernature.com as soon as possible.

Please note that your Supplementary Information files are now finalised, and they will be submitted as provided for preparation for publication of the Article. Any requests to make changes will only be considered in exceptional circumstances and will result in a delay to publication.

You will not receive access to your eProof until the Licence to Publish and Article-Processing Charge steps are completed.

We welcome the submission of material for the 'Featured Image' section of the Communications Materials home page. Images should relate to the content of your manuscript, but do not need to be taken directly from the accepted work. Suggestions should be sent by email to commsmat@nature.com, along with the completed <https://resource-cms.springernature.com/springer-cms/rest/v1/content/18943626/data/Research-Permission-Template-EN> Licence to Publish form. Please note that images should be supplied as 1400x400-pixel, in RGB. Unfortunately, we cannot promise that your suggestions will be used.

Providing great service is very important to us. We would greatly appreciate any comments you have about your experience at Communications Materials. We look forward to publishing your paper, and we hope to work with you again in the future.

Best regards,
John Plummer, PhD
Chief Editor
orcid.org/0000-0003-4824-8497
Communications Materials

*You can now use a single sign-on for all your accounts, view the status of all your manuscript submissions and reviews, access usage statistics for your published articles and download a record of your reviewing activity for the Nature Portfolio journals. Please check your account regularly and ensure that we have your current contact information.

We may promote your article on social media once it is published, so please feel free to send me the twitter handles of any authors or departments and we will be sure to tag them accordingly.

telephone: +90 312 210 26 28 fax: +90 312 210 26 00 e-mail: emrebuk@metu.edu.tr

December 25th, 2025

John Plummer, PhD
Chief Editor
Communications Materials

Manuscript ID: COMMSMAT-25-0901A
Title: Soft-Interfaced Liquid Crystal Microfluidics Can Probe the Rigidity of Lipid Vesicles
Author(s): Dedeoglu, Cansu; Bukusoglu, Emre

Dear Dr. Plummer,

We thank you for giving us chance to revise our manuscript based on the feedback provided by the Reviewers. We are grateful to the Reviewers for their helpful suggestions and comments, which helped us improve our manuscript. We thoroughly examined the comments and present our responses in detail under each comment. We also provided the revised manuscript that includes highlighted sections to show these changes.

Reviewer #1

The authors in the present article study the mechanical properties of lipid vesicles. The manuscript is unclear and confusingly written. It is difficult for the reader to understand the logical flowchart of the experiments and therefore their purpose. According to the authors, the purpose is to understand the role of cholesterol on the mechanical properties of phospholipid vesicles. Reading the experimental section, it becomes apparent that the authors do not use vesicles but an undefined phospholipid suspension. Vesicles are prepared from a phospholipid film in a test tube obtained by the evaporation of a solution containing the lipid and an organic solvent. Then, one proceeds with hydration using the buffer, followed by vortexing at a temperature higher than the gel-to-liquid crystal transition temperature of the lipid used. The vortexing step is the most important because it provides the necessary energy for bilayer formation. Systems containing cholesterol must be prepared in the same way by adding the programmed amount of cholesterol to the organic phase containing the phospholipid. The manuscript in its present form is not suitable for publication.

Response: We appreciate the Reviewer for their helpful criticism. We want to emphasize that the novelty of our work is the use of a microfluidic flow system with stabilized LC-aqueous interfaces that allow consistent, spatiotemporal quantification of the lipid adsorption (vesicle fusion)-induced optically observable changes at the LC-aqueous interfaces. With this capability, we show that we were able to track the minor mechanical alterations in lipid bilayers of the vesicles formed by a range of phospholipids and their mixtures through their spatiotemporal interfacial distribution at LC interfaces, which have not been investigated in detail previously. Considering the importance of the measurements of such properties in personal health tracking, which may prevent a potential cell dysfunction or critical diseases, we believe our work is of a significant interest in a broad community. We also concur that making the experimental methodology clear is crucial to conveying the goal and reasoning of the study. The Reviewer expressed concern that vesicles were not prepared following standard protocols and that the description may have suggested an undefined phospholipid suspension. We would like to point out that our vesicle preparation procedure indeed follows the established film hydration method as outlined by the reviewer and the literature. We first evaporate the chloroform containing the lipid, then hydrate it with buffer (PBS), and vortexing is performed to supply the necessary energy to form a bilayer. Moreover, for the cholesterol and lipid mixtures, cholesterol was prepared in the procedure described by the Reviewer. In addition, we provided diffusivity measurements of the lipid vesicles formed in aqueous suspensions in Fig. S13, showing consistent size measurements providing evidence on the formation of the lipid vesicles. We also provided results with three independent systems, whose results were correlated as described in the manuscript. We believe this clarification will resolve criticism raised by the Reviewer. We also expect that the Manuscript is improved in content and arguments after this round of revision.

Reviewer #2

The paper deals with an interesting and potentially important problem in the scarcely studied “no man’s land” between thermotropic liquid crystals (widely used, e.g., in electrooptical displays) and phospholipid membranes (which are among fundamental structures in cell biology). The lateral organization of the water-dispersed phospholipids (which is, in fact, liquid crystalline) at the interface between thermotropic liquid crystal and aqueous medium induces interface-driven orientational transitions of the thermotropic LC, in this particular case, the nematic 5CB. These transitions were studied in both “stagnant” films (the meaning is clear, but it is a rather strange term!), droplets, and microfluidic flow systems to record the response characteristics of the LC phases of 5CB upon fusion of lipid vesicles at the interface. The approach used in this paper is closely related to a series of works where the interface between 5CB and aqueous dispersions of various organic systems, mainly of surfactant nature, was used as an analytical tool providing qualitative (or even quantitative) information on the composition of the aqueous dispersion.

The general level of the paper is rather high, the results obtained for the microfluidic flow mode seem to be sufficiently new, and the narration is thorough and comprehensive. However, certain remarks should be made.

Response: We thank the Reviewer for their thoughtful and encouraging evaluation of our work. We sincerely value the acknowledgement of the originality of our microfluidic approach and the scientific significance of investigating the lateral

organization of phospholipids at the interface between thermotropic LCs and aqueous phase. We are also grateful for the careful reading of our manuscript, the acknowledgment of the overall comprehensiveness of our work and their remarks to improve the content of the article. These remarks are greatly appreciated and motivate us to refine further and advance this line of research. Below, we provide our response to the specific comments raised by the Reviewer and hope that we sufficiently addressed every issue brought up and made the required additions and changes.

Comment/Suggestion #1: The authors use several types of phospholipid systems (saturated, as DPPC or DMPC, and non-saturated, as DOPC or SOPC. The authors speak of “bending rigidity” K_c as an important parameter affecting the observed properties. Probably there should be some correlation between bending rigidity and phase transition points on DSC thermograms of hydrated phospholipids (which are clearly different for these phospholipids). This aspect should be analyzed in the revised version. Special attention should be paid to the systems phospholipid + cholesterol, for which this question is especially interesting.

Response: We thank the reviewer for this insightful comment, highlighting the possible relation between K_c and the phase behavior of lipids. Instead of thermal analysis, we performed additional new characterizations with the oscillatory barrier mode of the Langmuir Trough to independently characterize the mechanical properties of the monolayers of the pure lipids and their mixtures with cholesterol. By analyzing the results, we included necessary additions in the manuscript:

The detailed experimental procedure for the Langmuir Blodgett experiments was added into the Experimental section of the manuscript. The text on page 27 reads;

“Characterization of the Phospholipid Monolayers at the Water-Air Interface
Langmuir monolayer experiments were performed using a Langmuir Trough system (KN 2001, Biolin Scientific, Finland) equipped with a platinum Wilhelmy plate (Espoo, Finland) with a wetted length of 39.24 mm and barriers. Prior to each experiment, the trough and barriers were cleaned with ethanol by brushing and rinsed with ultrapure water. The aqueous sub-phase was filled into the system and cleaned the interfaces thoroughly. Lipid or lipid-cholesterol mixtures, which were prepared in chloroform with a constant amount of lipid as 1 mg/ml, were carefully spread dropwise into the air-water interface using a glass microsyringe in an amount corresponding to the desired molecule area. After spreading, the monolayer was allowed to equilibrate for 15 minutes to ensure the chloroform evaporation. Following the equilibration, the monolayer was compressed at a constant barrier speed of 10 mm/min. The surface pressure was recorded as a function of mean molecular area to obtain isotherms. For the surface rheological properties, the oscillating barrier method was used. The target surface pressures were selected as 27.5, 22.5, 32.5, and 30 mN/m for DLPC, DOPC, DPPC, and Egg SM, respectively. After compressing the monolayer to the target surface pressure, sinusoidal area oscillations were applied at a frequency of 30 mHz. All data were analyzed using the KSV NIMA LB software (Biolin Scientific).”

Moreover, we expanded the manuscript to include complementary mechanical characterization by Langmuir Trough oscillatory rheology and included new figures in the Supplementary Information. The results reveal systematic, lipid-specific trends in

the storage (G') and loss (G'') modulus with increasing cholesterol content, providing insights into elasticity. These measurements do not directly provide the bending rigidity of bilayers, but they provide interfacial mechanical properties and relaxation characteristics that qualitatively support the trends inferred from the fusion kinetics. In particular, DLPC and DOPC monolayers showed a cholesterol-induced stiffening up to 50 mol% cholesterol, consistent with the LC flow experiments. We also note that, for DPPC and Egg SM, gradual monotonic responses were measured with the limited amount of cholesterol content, as higher cholesterol content caused limitations in the measurements.

We included the following figures in the SI;

Figure S12. The plots of (a) the Langmuir isotherm and (b) the oscillating surface pressure with time for DLPC.

Figure S13. The plots of surface elastic modulus (G') and surface loss modulus (G'') with cholesterol content for (a) DLPC, (b) DPPC, (c) Egg SM, and (d) DOPC.

Figure S14. The plots of surface elasticity with cholesterol content and sample plots for oscillating barrier experiments for (a) DLPC and DLPC with 25% cholesterol, (b) DPPC and pure DPPC, (c) Egg SM and with 50% cholesterol, and (d) DPPC with 75% cholesterol.

We have updated the Results section of the manuscript in light of the results obtained from oscillatory Langmuir Trough measurements. Text on page 18 now reads;

“... To further analyze this trend, we obtained surface rheological responses of monolayers formed by pure and cholesterol-doped phospholipids using oscillatory barrier mode Langmuir Trough experiments (Fig. S12). The storage (G') and loss (G'') moduli exhibit a measurable dependence on lipid composition, type, and cholesterol content in the mixture, indicating pronounced changes in viscoelastic behavior.⁷⁸ For DLPC monolayers, Langmuir trough measurements reveal a predominantly viscous, fluid-like behavior as indicated by a higher G'' than G' (Fig. S13a). Upon introduction of the cholesterol, G' increases and becomes comparable to G'' , demonstrating a shift in mechanical properties toward a more elastic interfacial response. Moreover, the magnitudes of the complex elastic moduli (Fig. S14a) suggested the trend we obtained via our microfluidic and droplet experiments (Fig. 6).”

(78) Miyoshi, T.; Kato, S. Detailed Analysis of the Surface Area and Elasticity in the Saturated 1,2-Diacylphosphatidylcholine/Cholesterol Binary Monolayer System. *Langmuir* 2015, 31 (33), 9086–9096. <https://doi.org/10.1021/acs.langmuir.5b01775>

“...Furthermore, Langmuir Trough oscillatory measurements reveal that DPPC monolayers exhibit an elastic interfacial response as indicated by a higher G' than G'' (Fig. S13b), consistently with the literature.^{78,82} G'' and the magnitude of the complex

modulus (Fig. S14b) further increases with the cholesterol content of 25%, demonstrating the pronounced stiffening of the monolayer supporting the trend obtained by fusion kinetics.”

(82) Bykov, A. G.; Noskov, B. A. Surface Dilatational Elasticity of Pulmonary Surfactant Solutions in a Wide Range of Surface Tensions. *Colloid Journal* 2018, 80 (5), 467–473. <https://doi.org/10.1134/S1061933X18050034>.

“...Egg SM monolayers exhibited elastic interfacial characteristics with the G' dominating over G'' ”, indicating the formation of a rigid monolayer (Fig S13c). Upon the cholesterol introduction, the elastic characteristic remained the same with the lower complex elastic modulus value, indicating a decrease in elasticity with the cholesterol for 25%. At the 50% cholesterol content, the monolayer still exhibited a predominantly elastic interfacial characteristic with a relatively higher elastic modulus (Fig. S14c).”

“...As shown in Fig. S13d, DOPC monolayers exhibited a viscoelastic interfacial response with G' and G'' being comparable. Upon increasing cholesterol content, both moduli increased, and the enhancement of G' is higher than that of G'' , indicating a stiffening effect of monolayers (Fig. S14d). For the 75% amount of cholesterol, values are still comparable with each other, but with a lower magnitude.”

We have also updated the Discussion part by using these results.

“...Complementary Langmuir Trough oscillatory measurements revealed lipid-specific and cholesterol-dependent monolayer elasticity that qualitatively supported the trends observed in vesicle rigidity and fusion kinetics. Although the measurements do not directly provide quantification of bilayer rigidity, they provided independent mechanical insights into cholesterol-induced alterations in the mechanical properties of the monolayers. ...”

Comment/Suggestion #2: The abstract should be re-written with substantial changes. In the present form, the scientific novelty of the paper remains obscure – is it the microfluidics as an addition to other method of studies of phospholipid-5CB interactions, or the main sense is in mechanical properties (rigidity) of different phospholipids?

Response: We thank the Reviewer for this constructive suggestion. The novelty of our work is the use of the microfluidic flow system with stabilized LC-aqueous interfaces that allow consistent, spatiotemporal quantification of the lipid adsorption (vesicle fusion)-induced changes at the LC-aqueous interfaces. We showed that, with this capability, we were able to track the minor alterations in the mechanical properties of lipid bilayers, which have not been investigated in detail previously. Considering the importance of the measurements of such properties, which may lead to cell dysfunction and critical diseases, we believe our work is of a significant interest in a broad community. In order to highlight these, we revised the abstract as follows.

“Liquid crystal (LC)-aqueous interfaces were shown to respond to the phospholipid interactions via optically observable ordering transitions; however, past attempts lack the quantification of the transport and fusion kinetics of the vesicles at the interfaces. Herein, we investigated the response of flowing LC-aqueous interfaces upon fusion of

the vesicles formed by pure 1,2-dilauroyl-sn-glycero-3-phosphocholine (DLPC), 1,2-dioleoyl-sn-glycero-3-phosphocholine (DOPC), 1,2-dipalmitoyl-sn-glycero-3-phosphocholine (DPPC), or egg sphingomyelin or their mixtures with guest molecules. Using stabilized LC-aqueous interfaces in transparent microfluidic chips that allow spatiotemporal quantification using fluorescence, confocal, and polarized light microscopy, we demonstrated that flowing LC interfaces provide a rapid response to lipid adsorption, where their spatiotemporal interfacial distribution differs depending on the mechanical properties of their vesicles. We show that cholesterol-dissolved lipid complexes result in distinct LC-response kinetics, mainly associated to the changes in their rigidities. Considering the critical role of the mechanical properties of cell membranes in proper cellular function, this study is significant as it offers a continuous and rapid early diagnosis platform for detecting minor mechanical alterations in lipid bilayers, which may lead to cell dysfunction that contributes to critical diseases.”

Comment/Suggestion #3: The general idea seems to have grown from the papers of Abbott e.a. published in 2005 and even earlier. The progress in this topic during recent 20 years (which is undeniable) should be clearly described.

Response: We thank the Reviewer for bringing this issue to our attention. We agree that the early foundational work by Abbott and colleagues in the early 2000s must be combined by progression studies, and they should be clearly acknowledged. In addition to the references cited in the original manuscript (refs. 26-32 in revised manuscript), we included additional literature in the Introduction section to provide a more comprehensive overview of these developments and to better situate our study within this historical context. The studies using lipid-decorated LC-aqueous interface advanced in several distinct directions, which we illustrate using representative studies from the literature. For example, the disruption of phospholipid monolayers at the 5CB-aqueous interface by protein-coated nanoparticles was shown to induce orientational transitions of LCs and thereby optical textures, depending on the protein type, and demonstrating that LC interfaces can probe potential cytotoxicity of nanomaterials (33). Complementary works revealed that LC ordering transitions can detect early-stage β -amyloid aggregation on lipid monolayers, highlighting the sensitivity of LC ordering to subtle changes in membrane-associated peptide organization (34). Recent advances in understanding biomolecular interactions based on disrupting the phospholipid monolayers at the LC-water interface provide probing cytoplasmic proteins that modulate surface anchoring and defect structures in predictable ways and mutation-dependent functional alterations in pore-forming toxins (35,36). Further advances in this field were achieved through the use of lipid-decorated LC droplets, where curvature allowed precise control of the LC director via synthetic phospholipids with different acyl chains and the nature of the LC, providing insights into protein interaction and how membrane-like assemblies impose boundary conditions on confined nematic phases (37, 38,39,40). Moreover, different systems using shells and smectic LC were utilized for characterizing the lateral organization of the phospholipids at the interfaces. (41, 42). This study provides a valuable progress from our starting point, as it demonstrates how LC droplet-based systems and variations in the nature of lipids can be exploited to systematically investigate interfacial organization and orientational responses of LC, motivating our exploration of lipid-based mixtures with different natures in our study. These revisions are now included in the Introduction section of the revised manuscript. The added text in the Introduction section:

“...Building on these early findings, studies showed that disruption of the lateral organization of phospholipid monolayers at the stagnant LC-aqueous interfaces by protein-coated nanoparticles³³ or amyloid-forming peptides generates distinct changes in LC orientational ordering, enabling label-free detection and real-time response.³⁴ Subsequent studies showed that lipid-coated LC-aqueous interface can be directly utilized for probing protein-lipid interactions in a similar manner, allowing the highly sensitive detection of cytoplasmic proteins³⁵ and mutation-dependent activity of toxins, highlighting the importance of studying biomolecular interactions at the interface.³⁶ Building on this approach, LC droplets have also been used to investigate nanoscale lipid–protein interactions, allowing for quantitative analysis of lipid–protein coupling at submicrometer scales by producing orientational transitions.^{37,38} In addition, lipid-coated LC droplets in aqueous media emerged as a platform for controlling director fields through the organization at the interface, further highlighting the characterization of lipids.^{39,40} In parallel, phospholipid-coated LC shells were demonstrated to create stable lipid islands whose spatial organization induces a change in the director inside the shell, demonstrating how confinement and curvature can enhance interfacial lipid heterogeneity.⁴¹ Moreover, lateral heterogeneity and redistribution of the lipids at the interface can be caused by LC phase transitions themselves, as demonstrated by the local concentration of phospholipids into high-density regions caused by smectic LC nucleation at aqueous interfaces.⁴²”

(33) Hartono, D., Qin, W. J., Yang, K. L., & Yung, L. Y. L. (2009). Imaging the disruption of phospholipid monolayer by protein-coated nanoparticles using ordering transitions of liquid crystals. *Biomaterials*, 30(5), 843–849. <https://doi.org/10.1016/j.biomaterials.2008.10.037>

(34) Sadati, M., Apik, A. I., Armas-Perez, J. C., Martinez-Gonzalez, J., Hernandez-Ortiz, J. P., Abbott, N. L., & de Pablo, J. J. (2015). Liquid Crystal Enabled Early Stage Detection of Beta Amyloid Formation on Lipid Monolayers. *Advanced Functional Materials*, 25(38), 6050–6060. <https://doi.org/10.1002/adfm.201502830>

(35) Devi, M., Verma, I., & Pal, S. K. (2021). Distinct interfacial ordering of liquid crystals observed by protein-lipid interactions that enabled the label-free sensing of cytoplasmic protein at the liquid crystal-aqueous interface. *Analyst*, 146(23), 7152–7159. <https://doi.org/10.1039/d1an01444g>

(36) Gupta, T., Lata, K., Chattopadhyay, K., & Pal, S. K. (2025). Utilizing an aqueous-liquid crystal interface to investigate membrane protein interactions and mutation effects of a pore-forming toxin. *Journal of Materials Chemistry B*, 13(18), 5358–5364. <https://doi.org/10.1039/d4tb02117g>

(37) Pani, I., Fidha Nazreen, K. M., Sharma, M., & Pal, S. K. (2021). Probing Nanoscale Lipid-Protein Interactions at the Interface of Liquid Crystal Droplets. *Nano Letters*, 21(11), 4546–4553. <https://doi.org/10.1021/acs.nanolett.0c05139>

(38) Lin, I. H., Miller, D. S., Bertics, P. J., Murphy, C. J., de Pablo, J. J., & Abbott, N. L. (2011). Endotoxin-induced structural transformations in liquid crystalline droplets. *Science*, 332(6035), 1297-1300.

(39) Paterson, D. A., Bao, P., Abou-Saleh, R. H., Peyman, S. A., Jones, J. C., Sandoe, J. A. T., Evans, S. D., Gleeson, H. F., & Bushby, R. J. (2020). Control of Director Fields in Phospholipid-Coated Liquid Crystal Droplets. *Langmuir*, 36(23), 6436–6446. <https://doi.org/10.1021/acs.langmuir.0c00651>

(40) Carter, M. C., Miller, D. S., Jennings, J., Wang, X., Mahanthappa, M. K., Abbott, N. L., & Lynn, D. M. (2015). Synthetic mimics of bacterial lipid A trigger optical transitions in liquid crystal microdroplets at ultralow picogram-per-milliliter concentrations. *Langmuir*, *31*(47), 12850-12855.

(41) Sharma, A., Gupta, D., Scalia, G., & Lagerwall, J. P. F. (2022). Lipid islands on liquid crystal shells. *Physical Review Research*, *4*(1).
<https://doi.org/10.1103/PhysRevResearch.4.013130>

(42) Yang, C., Chen, L., Zhang, R., Chen, D., Arriaga, L. R., & Weitz, D. A. (2022). Local high-density distributions of phospholipids induced by the nucleation and growth of smectic liquid crystals at the interface. *Chinese Chemical Letters*, *33*(8), 3973–3976.
<https://doi.org/10.1016/j.ccl.2021.11.016>

Comment/Suggestion #4: A similar approach has been widely used for another problem – absorption or organics from water dispersions on oriented 5CB structures as means for optical sensing in biomedical applications (there are plenty of papers using similar methods, but without any mentioning of phospholipids). Again, the question – the place of this work among multiple analogues should be clearly outlined.

Response: We thank the Reviewer for this insightful and valuable suggestion. The literature reveals that LC-based optical sensing platforms have been extensively used to detect small amounts of various organic species in aqueous media without involving phospholipids. To clarify the position of our work within this broader literature, we have revised the manuscript. Specifically, we presented references on representative sensing aimed at LC studies. This field has indeed expanded with various studies demonstrating that cholesteric and nematic LC droplets can act as a sensing platform for a wide range of chemical and biological events. Cholesteric droplets showed high sensitivity and fast response for biologically relevant dispersions such as glucose, hemoglobin, and cholesterol (43), while complex LC emulsions have been engineered to enable highly sensitive detection of boronic acid polymeric surfactants caused by reversible interactions with antibodies. (44) Parallel advances have shown that the LC-aqueous interface can be modulated not only by direct adsorption of molecules but also by reaction products in aqueous media, such as α -glucosidase to inhibit anti-diabetic drugs. (45) Moreover, LC-based aptasensors were successfully applied to detect malathion with a very low detection limit by using optical changes of the interface. (46) Additional studies demonstrated that LC-based optical devices can be used for sensing kanamycin by utilizing enzyme-linked dual-function nucleic acid on magnetic beads. (47) Moreover, using synthetic enzyme mimics, including fullerene-based catalysts, influenced LC responses by enzymatic hydrolysis, promising use in biochemical sensing and diagnostic technologies. (48,49) Among this rich literature in LC-based optical sensing, our work interrogates intrinsic physicochemical and mechanical properties of lipid vesicles themselves, such as with defined cholesterol content, through their spatiotemporal fusion characteristics at the interface. Additionally, our work integrates a novel microfluidic platform forming a soft flowing interface between LC and aqueous phase, enabling real-time observation and faster response.

The added text in the Introduction section:

“...In addition to lipids and amphiphilic biomolecular interactions-mediated ordering transitions, LC-aqueous interfaces have been extensively used as optical biosensing platforms, where adsorption of biological species such as glucose⁴³, polymeric surfactants⁴⁴, or reaction-generated products such as α -glucosidase to inhibit anti-diabetic drugs⁴⁵ induces changes in orientational order. These interfaces have also emerged in aptamer-functionalized systems that detect pesticides⁴⁶ and nucleic acid-linked devices, enabling antibiotic sensing.⁴⁷ Recent studies revealed that synthetic enzyme mimics, such as fullerene-based catalytic assemblies, can induce a change in the optical appearance of LC, further highlighting LC platforms in biosensing applications.^{48,49}”

(43) Lee, H. G., Munir, S., & Park, S. Y. (2016). Cholesteric Liquid Crystal Droplets for Biosensors. *ACS Applied Materials and Interfaces*, 8(39), 26407–26417. <https://doi.org/10.1021/acsami.6b09624>

(44) Concellón, A., Fong, D., & Swager, T. M. (2021). Complex Liquid Crystal Emulsions for Biosensing. *Journal of the American Chemical Society*, 143(24), 9177–9182. <https://doi.org/10.1021/jacs.1c04115>

(45) Sun, H., Yin, F., Liu, X., Jiang, T., Ma, Y., Gao, G., Shi, J., & Hu, Q. (2021). Development of a liquid crystal-based α -glucosidase assay to detect anti-diabetic drugs. *Microchemical Journal*, 167. <https://doi.org/10.1016/j.microc.2021.106323>

(46) Nguyen, D. K., & Jang, C. H. (2021). A cationic surfactant-decorated liquid crystal-based aptasensor for label-free detection of malathion pesticides in environmental samples. *Biosensors*, 11(3). <https://doi.org/10.3390/bios11030092>

(47) Song, H., Khan, M., Yu, L., Wang, Y., Lin, J. M., & Hu, Q. (2023). Construction of Liquid Crystal-Based Sensors Using Enzyme-Linked Dual-Functional Nucleic Acid on Magnetic Beads. *Analytical Chemistry*, 95(35), 13385–13390. <https://doi.org/10.1021/acs.analchem.3c03163>

(48) Karaman, D., Akar, E., Saylam, A., Özçubukçu, S., & Bukusoglu, E. (2025). Engineered Fullerene-Based Enzyme Mimics for Enhanced Sensing of Aqueous-Phase Reactions Using Liquid Crystals. *ACS Applied Nano Materials*, 8(47), 22798–22808. <https://doi.org/10.1021/acsnm.5c04258>

(49) Karaman, D., Saylam, A., Akar, E., Özçubukçu, S., & Bukusoglu, E. (2025). Liquid Crystals that Respond to the Aqueous Phase Reactions Catalyzed by Synthetic Enzyme Mimics. *Advanced Materials Interfaces*. <https://doi.org/10.1002/admi.202500531>

We also added the text in the following to the Discussion section to clarify the place of this work in the literature.

“...Recently, LC-aqueous interfaces have emerged as optical biosensing platforms where biomolecular interactions such as analyte binding and enzymatic reactions induce a change in the optical appearance of the interface. This system could find a niche in these studies by aiming to investigate intrinsic mechanical properties of membranes rather than external biochemical species and interactions.”

We hope after these revisions, our manuscript is ready for publication. We thank you for your handling of our manuscript.

Sincerely,

Emre Bukusoglu
Associate Professor
Department of Chemical Engineering
Middle East Technical University

February 5th, 2026

John Plummer, PhD
Chief Editor
Communications Materials

Manuscript ID: COMMSMAT-25-0901A

Title: Soft-Interfaced Liquid Crystal Microfluidics Can Probe the Rigidity of Lipid Vesicles

Author(s): Dedeoglu, Cansu; Bukusoglu, Emre

Dear Dr. Plummer,

We thank you for giving us a chance to revise our manuscript based on the feedback provided by the reviewers. We present our responses in detail below. We also provide the revised manuscript that includes highlighted sections to show these changes.

Reviewer #1

In their rebuttal letter to the Editor, the authors state the following: “We would like to point out that our vesicle preparation procedure indeed follows the established film hydration method as outlined by the reviewer and the literature. We first evaporate the chloroform containing the lipid, then hydrate it with buffer (PBS), and vortexing is performed to supply the necessary energy to form a bilayer.” However, in both the marked and unmarked versions of the main manuscript, the authors write: “To prepare a lipid solution for the experiments, 0.5 μ L of DLPC (25 mg/mL in chloroform) was taken into the glass vial. The gentle flow of the nitrogen stream was used for evaporating the chloroform. 4 mL PBS solution was added to achieve a 5.0 μ M DLPC solution. This solution was diluted with a PBS solution to the desired concentration used in experiments. The same procedure was applied to other lipids.” Based on this text, there is no mention of vortexing, the preparation temperature, or the use of a vacuum pump for chloroform removal. I cannot verify whether the authors' claims in the letter to the Editor are accurate. As it stands, the experimental description indicates that the prepared sample does not consist of vesicles, but rather an undefined phospholipid suspension.

Furthermore, the authors should be aware that even with vortexing, one typically obtains multilamellar vesicles (MLVs), which have an "onion-like" structure and a large radius of curvature that significantly influences vesicle fusion phenomena. While small vesicles have a strong propensity to fuse, large ones do not. Therefore, reporting the size of the vesicles used is crucial.

I would like to remind the authors that an accurate description of sample preparation is fundamental to the integrity of a manuscript. An experiment must be reproducible in any laboratory; however, without a detailed description of the experimental conditions, replication is impossible.

Response: We appreciate the Reviewer for their criticism and agree that the full description of the methods and experimental work is fundamental for a scientific publication. We went through our experimental methods and revised the text accordingly in the revised version of the manuscript. We also provided size measurements of the vesicles resulting in this procedure along with the diffusivity measurements reported in Figure S11. The revised text is shown below.

The text on page 26 (Experimental Section, Preparation of Vesicle Suspensions) now reads;

“... To prepare a lipid solution for the experiments, 0.5 μL of DLPC (25 mg/mL in chloroform) was taken into the glass vial. The gentle flow of the nitrogen stream was used for evaporating the chloroform. After further evaporation of the residual chloroform with vacuum (< 50 mbar) for 5 minutes at 60°C , 4 mL PBS solution was added to achieve a $5.0 \mu\text{M}$ DLPC solution and vortexed at 3000 rpm for 5 mins for the dispersion of vesicles in suspension. This solution was diluted with a PBS solution to the desired concentration to be used in experiments. The same procedure was applied to other lipids; however, vortex mixing was performed at room temperature for DLPC and DOPC whereas was performed at 50°C for DPPC and Egg SM. ...”

On page 27;

“... . The diffusion coefficients and average sizes are reported in Figure S11 as mean \pm standard deviation of the three independent measurements. ...”

While revising the text, we realized it would be better to provide additional details about the microfluidic coating procedures. For this purpose, we revised the text on page 26 as follows;

“To achieve the stable two-phase flow, DMOAP (from a 1% volume solution in water) was coated to approximately one-half of the channel in its longitudinal direction using flow focusing using 1% DMOAP solution and deionized water co-flow system. The inlet pressure of the DMOAP solution was set to 190 mbar, and 200 mbar was set for water. The simultaneous flows were ensured for 30 minutes. The channel was initially filled with water flow and DMOAP solution was introduced with a 3 minute ramp. “

The revised Figure S11 and the caption in the supplementary information is as follows;

a) DLPC

b) DPPC

c) Egg SM

d) DOPC

Figure S11. The plots of diffusion coefficients (left y-axis) and size (right y-axis) of the vesicles formed by (a) DLPC, (b) DPPC, (c) Egg SM, (d) DOPC with cholesterol sketched vs. cholesterol content (%).

Reviewer #2

I think that the authors have given satisfactory answers to my questions and remarks. The revised text has been substantially improved, and now I think that the paper should be accepted for publication.

Response: We thank the Reviewer for their evaluation and support in the publication of our work.

We hope that these revisions make our manuscript ready for publication. We appreciate your handling of our manuscript.

Sincerely,

Emre Bukusoglu
Associate Professor
Department of Chemical Engineering
Middle East Technical University